# Zinc activation of OTOP proton channels identifies structural elements of the gating apparatus

**Bochuan Teng[1,2†], Joshua P Kaplan[1,2†], Ziyu Liang[1,2], Kevin Saejin Chyung[1], Marcel P Goldschen-Ohm[3], Emily R Liman[1,2]\***

[1]Section of Neurobiology, Department of Biological Sciences, University of Southern California, Los Angeles, United States; [2]Program in Neuroscience, University of Southern California, Los Angeles, United States; [3]University of Texas at Austin, Department of Neuroscience, Austin, United States

**Abstract** Otopetrin proteins (OTOPs) form proton-selective ion channels that are expressed in diverse cell types where they mediate detection of acids or regulation of pH. In vertebrates there are three family members: OTOP1 is required for formation of otoconia in the vestibular system and it forms the receptor for sour taste, while the functions of OTOP2 and OTOP3 are not yet known. Importantly, the gating mechanisms of any of the OTOP channels are not well understood. Here, we show that zinc ($Zn^{2+}$), as well as other transition metals including copper ($Cu^{2+}$), potently activates murine OTOP3 (mOTOP3). $Zn^{2+}$ pre-exposure increases the magnitude of mOTOP3 currents to a subsequent acid stimulus by as much as 10-fold. In contrast, mOTOP2 currents are insensitive to activation by $Zn^{2+}$. Swapping the extracellular tm 11–12 linker between mOTOP3 and mOTOP2 was sufficient to eliminate $Zn^{2+}$ activation of mOTOP3 and confer $Zn^{2+}$ activation on mOTOP2. Mutation to alanine of H531 and E535 within the tm 11–12 linker and H234 and E238 within the 5–6 linker reduced or eliminated activation of mOTOP3 by $Zn^{2+}$, indicating that these residues likely contribute to the $Zn^{2+}$ activating site. Kinetic modeling of the data is consistent with $Zn^{2+}$ stabilizing the opn2+en state of the channel, competing with $H^+$ for activation of the channels. These results establish the tm 11–12 and tm 5–6 linkers as part of the gating apparatus of OTOP channels and a target for drug discovery. $Zn^{2+}$ is an essential micronutrient and its activation of OTOP channels will undoubtedly have important physiological sequelae.

**\*For correspondence:**
liman@usc.edu

†These authors contributed equally to this work

## Editor's evaluation

This valuable study discovers that zinc ions can activate some OTOP proton channels, identifying a pharmacological tool for research, and further establishing that OTOP channels gate. The data presented provides convincing support for conclusions. This work is expected to be of great interest to physiologists studying OTOP channels and other proton-permeation pathways.

## Introduction

Pharmacological agents that can activate or inhibit ion channels have long been used as probes to describe the fundamental processes of channel gating and ion permeation (*Hille, 2001*). For example, the discovery of the charged molecule TEA and the scorpion toxin charybdotoxin as a specific blocker of $K^+$ channels allowed for the early identification of residues lining the channel pore well before the channel structures were determined (*MacKinnon et al., 1990*; *Yellen et al., 1991*; *Banerjee et al., 2013*). Similarly, gating modifiers have been used to probe structural rearrangements that

accompany the opening of voltage-gated ion channels (*Swartz and MacKinnon, 1997*; *Sack and Aldrich, 2006*; *Catterall et al., 2007*; *Goldschen-Ohm and Chanda, 2014*). More recently, toxins that target pain-sensing ASIC and TRPV1 channels have been used to probe the conformational states of these channels (*Bohlen et al., 2010*; *Baconguis et al., 2014*). One of the most common modulators of channel activity is the trace metal zinc ($Zn^{2+}$), which can affect gating, permeation, or both (*Gilly and Armstrong, 1982*; *Chu et al., 2004*; *Noh et al., 2015*; *Peralta and Huidobro-Toro, 2016*). $Zn^{2+}$ binds to proteins with high affinity and specificity and regulates a wide range of cellular processes, including metabolism and gene expression (*Vallee and Falchuk, 1993*). $Zn^{2+}$ is a potent inhibitor of proton transport molecules including the voltage-gated proton channel Hv1 and the proton-selective ion channel OTOP1 (*Decoursey, 2003*; *Ramsey et al., 2006*; *Bushman et al., 2015*; *Tu et al., 2018*; *Teng et al., 2019*).

OTOP1 is a member of a family of proteins (*Hughes et al., 2008*; *Hurle et al., 2011*), which includes, within vertebrates, two other members, OTOP2 and OTOP3, that also function as proton channels (*Tu et al., 2018*). OTOP proton channels are expressed throughout the body, where they play diverse and still poorly understood roles in pH sensing and homeostasis. In vertebrates and invertebrates, OTOP channels expressed in the gustatory system sense acids and function as sour taste receptors (OTOP1 for vertebrates; OTOPL1 for *Drosophila*) (*Teng et al., 2019*; *Zhang et al., 2019*; *Ganguly et al., 2021*; *Mi et al., 2021*). In mice and zebrafish, OTOP1 plays an essential role in the formation of force-sensing calcium carbonate-based otoconia in the ear (*Hurle et al., 2003*; *Hughes et al., 2004*), likely by regulating pH in the endolymph. OTOP2 and OTOP3 are both found throughout the digestive system, and their expression has been shown to correlate with disease progression in some forms of colon cancer (*Tu et al., 2018*; *Parikh et al., 2019*; *Yang et al., 2019*). Most recently, an OTOP channel was shown to be critically involved in calcification and the formation of a skeleton in sea urchin embryos (*Chang et al., 2021*).

Given the recent discovery of OTOP proteins as forming ion channels (*Tu et al., 2018*), much remains to be discovered about how they function. For example, it was not known if the channels occupy open and closed states or if those terms even apply to these proteins, which bear no structural similarity to other ion channels, and that could conduct protons through a non-aqueous pathway (*Decoursey, 2003*). Recently, we showed that OTOP channels are gated by extracellular protons, acting mostly likely on multiple titratable residues on the extracellular domain of the protein (*Teng et al., 2022*). Here, we report the first evidence that OTOP channels can be activated by $Zn^{2+}$ and $Cu^{2+}$. This confirms that OTOP channels are gated, like nearly all other ion channels. Using a chimeric channel approach and point mutations, we identify residues in the linkers between tm 11-12 and tm 5-6 that represent key determinants for and likely form the $Zn^{2+}$ activating site. We further propose that the elements of the gating apparatus of the channels that we identify are potential targets for pharmacological manipulation.

## Results

### $Zn^{2+}$ both blocks and potentiates OTOP3 currents

While measuring the sensitivity of the three murine OTOP channels to inhibition by $Zn^{2+}$, we noticed that mOTOP3 currents were larger following removal of $Zn^{2+}$ than before its introduction. We refer to this activating effect of $Zn^{2+}$ as potentiation. For these and other experiments, murine OTOP channels were expressed in HEK-293 cells and studied by patch-clamp recording. As shown in *Figure 1*, all three murine OTOP channels carried inward currents in response to a pH 5.5 stimulus and were subsequently inhibited by 1 mM $Zn^{2+}$ at pH 5.5. However, only mOTOP3 currents showed recovery during the $Zn^{2+}$ exposure and a large rebound, of nearly threefold, following its removal (*Figure 1A and B*). The potentiating effect of $Zn^{2+}$ was dose-dependent over a concentration range of 0.1–10 mM and did not show evidence of saturation (*Figure 1C and D*). This contrasts with the inhibitory effect of $Zn^{2+}$ on mOTOP3 currents, which for a stimulus of pH 5.5 showed clear saturation at 3 mM and could be fit with an $IC_{50}=0.31$ mM (*Figure 1D*). The difference in the dose dependence of inhibition and potentiation suggests that $Zn^{2+}$ interacts with distinct binding sites on the channels to produce the two effects (see below).

We next asked whether the effect of $Zn^{2+}$ to potentiate the mOTOP3 currents was through actions on the extracellular or the intracellular side of the channel, possibly following entry into the cytosol as

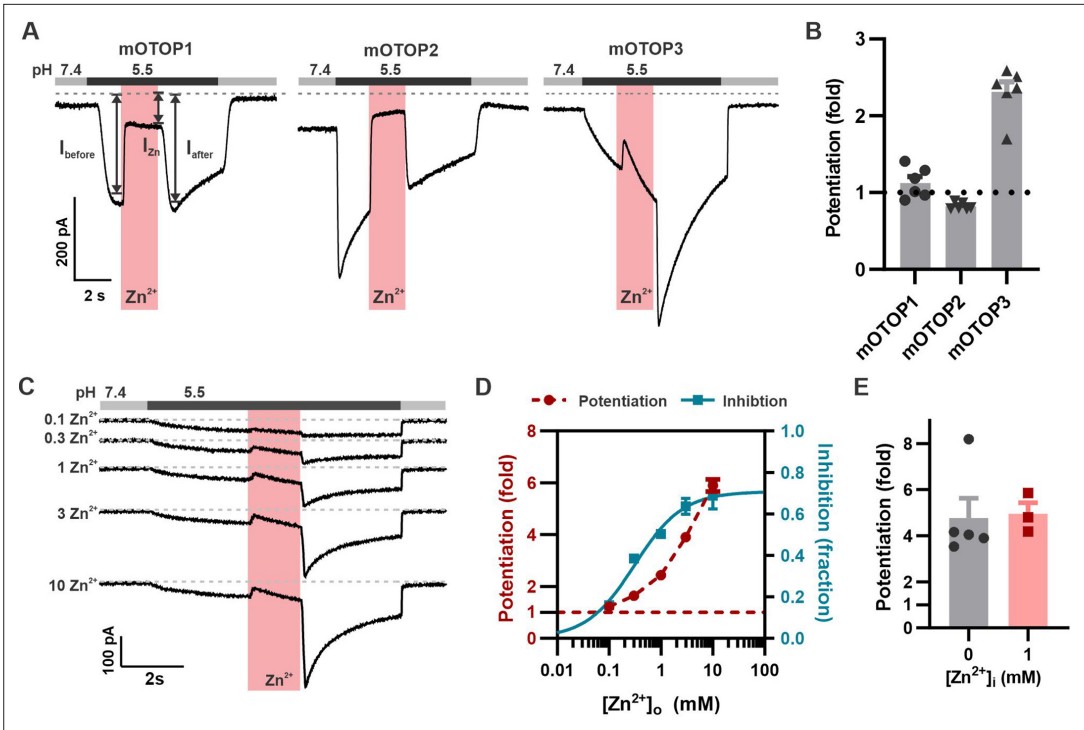

**Figure 1.** $Zn^{2+}$ blocks and potentiates mOTOP3 currents. (**A**) Representative traces show $Zn^{2+}$ both inhibits and potentiates mOTOP3 currents. Proton currents were elicited in HEK293 cells expressing each of the three mOTOP channels in response to lowering the extracellular pH to 5.5 in the absence of extracellular $Na^+$ as indicated. $Zn^{2+}$ (pink bar, 1 mM) inhibits currents through all three channels, but only mOTOP3 currents are potentiated following $Zn^{2+}$ removal. $V_m$ was held at –80 mV. (**B**) Average and all points data for experiments as in (**A**) showing fold potentiation as measured by comparing the current magnitude before and after $Zn^{2+}$ application (arrows shown in **A**) .(**C**) $Zn^{2+}$ applied at varying concentrations (pink bar, concentration indicated in mM) produces a dose-dependent inhibition and potentiation of mOTOP3 currents. (**D**) Average data from experiments in (C) show the dose dependence of potentiation and inhibition (n=4 for $Zn^{2+}$ potentiation, n=6 for $Zn^{2+}$ inhibition). The dose dependence of $Zn^{2+}$ inhibition was fit with a Hill equation, with an $IC_{50}$=0.31 mM, and Hill coefficient = 0.94. (**E**) Average potentiation of mOTOP3 currents in response to 1 mM extracellular $Zn^{2+}$ with (gray) or without (pink) 1 mM $Zn^{2+}$ loaded in the pipette. There was no difference between the two conditions (Student's t-test, p=0.88).

The online version of this article includes the following source data for figure 1:

**Source data 1.** Source data for **Figure 1**.

is the case for TRPA1 (**Hu et al., 2009**). Introducing 1 mM $Zn^{2+}$ into the patch pipette did not change the degree of potentiation in response to 1 mM extracellular $Zn^{2+}$, indicating that $Zn^{2+}$ likely acts on extracellular domains of the channel (**Figure 1E**).

## Pre-exposure to $Zn^{2+}$ potentiates OTOP3 currents

To study the activating effects of $Zn^{2+}$ on mOTOP3 and avoid confounds due to its inhibitory effect, we devised a recording protocol in which the cells were pre-exposed to $Zn^{2+}$ at pH 7.4, prior to evoking currents with an acidic stimulus (pH 5.5). As shown in **Figure 2A**, this produced a robust potentiation of mOTOP3 currents evoked in response to the pH 5.5 stimulus. Notably, the currents were both faster and larger after exposure to $Zn^{2+}$.

We measured the dose and time dependence of potentiation by $Zn^{2+}$, using three concentrations of $Zn^{2+}$: 0.3, 1, and 3 mM and by varying the duration of the $Zn^{2+}$ pre-exposure from 1 to 64 s. The response to $Zn^{2+}$ was compared to the response in the absence of $Zn^{2+}$ from the same cell. As shown in **Figure 2B and C**, 3 mM $Zn^{2+}$ caused a more than 10-fold increase in the peak current, the lowest concentration of $Zn^{2+}$, 0.3 mM, applied for up to 64 s produced only a negligible increase in the peak current and 1 mM $Zn^{2+}$ had an intermediate effect. Thus, the potentiating effect of $Zn^{2+}$, as measured by the peak current, was dose-dependent, with an apparent threshold of >0.3 mM $Zn^{2+}$. Examination

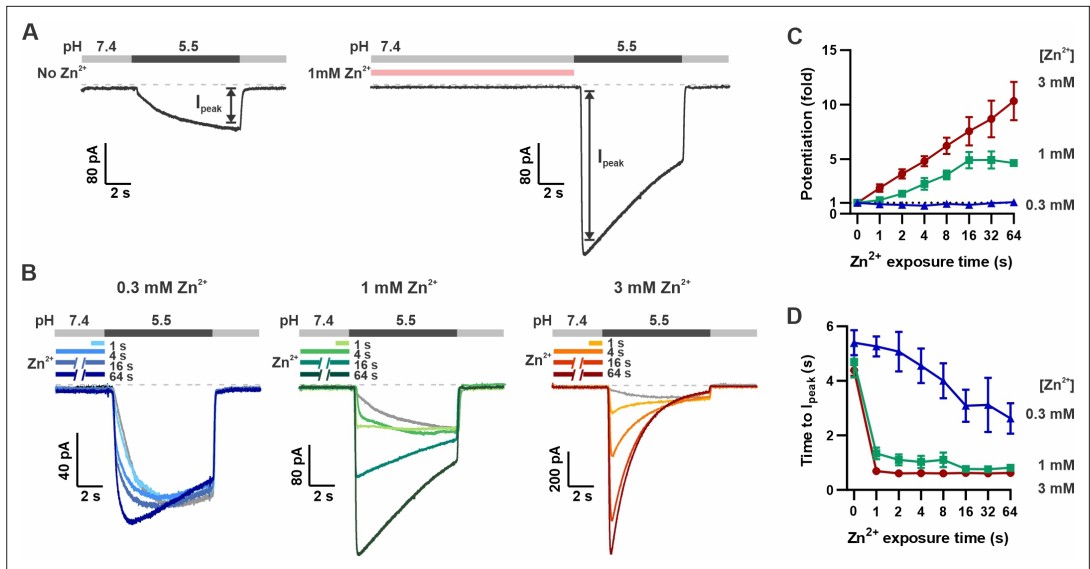

**Figure 2.** Pre-exposure to $Zn^{2+}$ potentiates mOTOP3 currents in a dose- and time-dependent manner. (**A**) Solution exchange protocol designed to measure effects of $Zn^{2+}$ on gating of OTOP currents without confounds due to its inhibitory effects. $V_m$ was held at –80 mV. In this example, currents were elicited to a pH 5.5 stimulus without pre-exposure to $Zn^{2+}$ and then following a 16 s exposure to 1 mM $Zn^{2+}$. (**B**) Representative traces show mOTOP3 currents elicited in response to pH 5.5 stimulus with pre-exposure to 0.3 mM (blue), 1 mM (green), and 3 mM (orange/red) $Zn^{2+}$ for durations from 1 to 64 s as indicated. (**C, D**) The fold potentiation (**C**) and time to $I_{peak}$ (**D**) as a function of $Zn^{2+}$ pre-exposure time from experiments as in (B) (n=5–7). Fold potentiation was measured as the ratio of the current evoked to the pH 5.5 stimulus after $Zn^{2+}$ to the control response in the absence of $Zn^{2+}$. Data are plotted as mean ± s.e.m.

The online version of this article includes the following source data for figure 2:

**Source data 1.** Source data for *Figure 2*.

of the time dependence of the response showed increasing potentiation with exposure times up to ~16 s for concentrations of 1 and 3 mM $Zn^{2+}$, at which point the effect tended to saturate, although there was some variability from cell to cell (*Figure 2C*).

In addition to increasing the peak current, $Zn^{2+}$ pre-exposure also increased the apparent rate of activation to a pH 5.5 stimulus that otherwise slowly activates mOTOP3 currents (*Teng et al., 2022*). This change in apparent activation kinetics showed a dose and time dependence (*Figure 2D*). At the higher concentrations (1 and 3 mM), an exposure of 1 s was sufficient to observe a maximal decrease in the time to peak current from ~4 s to <1 s. At 0.3 mM $Zn^{2+}$, this effect required longer exposure times, and activation rates never reached the speed obtained with 1 s exposure to 1 mM $Zn^{2+}$. To assess the stability of the $Zn^{2+}$ bound conformation, we measured the rate of recovery from potentiation by $Zn^{2+}$ (*Figure 3*). For these experiments, the cells were exposed to 1 mM $Zn^{2+}$ (pH 7.4) for 16 s. This was followed by a 'wash-off' period of 0–64 s in a $Zn^{2+}$-free solution (pH 7.4) before currents were activated with a pH 5.5 solution (*Figure 3A*). A wash-off period of 1 s was sufficient to reduce potentiation by 50% while a period of >16 s allowed for a complete recovery of currents to baseline.

Thus, application of millimolar concentrations of $Zn^{2+}$ at pH 7.4 elicits upon its removal a robust concentration- and time-dependent potentiation of mOTOP3 currents.

## OTOP1 is mildly potentiated by $Zn^{2+}$

Using this new protocol, we went back and assessed the effect of $Zn^{2+}$ on mOTOP1 and mOTOP2. mOTOP1 and mOTOP2 currents evoked in response to a solution at pH 5.5 showed little to no evidence of potentiation by exposure to 1 mM $Zn^{2+}$ applied for 16 s at pH 7.4 (*Figure 4A and B*). As mOTOP1 and mOTOP2 differ from mOTOP3 channels in that they display a greater degree of activation at pH 5.5 (*Teng et al., 2022*), we considered whether this might preclude further potentiation by $Zn^{2+}$. Indeed, we found that mOTOP1 currents evoked in response to a pH 6.0 stimulus could be potentiated by as much as threefold after a 64 s exposure to 1 mM $Zn^{2+}$ (*Figure 4C*), with a time

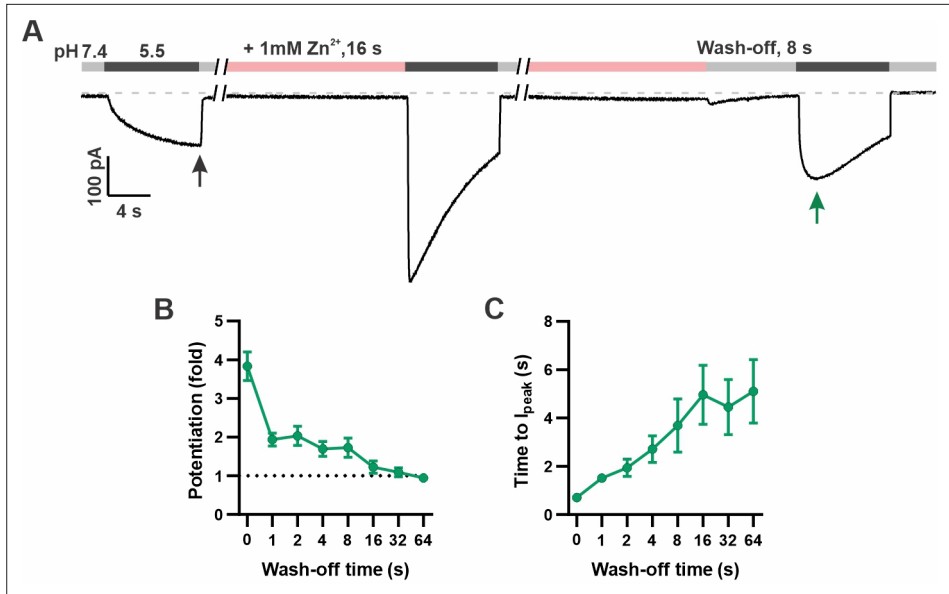

**Figure 3.** Time dependence of the recovery of mOTOP3 currents from $Zn^{2+}$ pre-potentiation. (**A**) Solution exchange protocol designed to measure the recovery of mOTOP3 currents following exposure to $Zn^{2+}$. In this example, the cell expressing mOTOP3 was first exposed to 1 mM $Zn^{2+}$ for 16 s which was followed by an 8 s wash-off phase in pH 7.4 solution before currents were elicited in response to the pH 5.5 solution. (**B, C**) The fold potentiation (**B**) and time to $I_{peak}$ (**C**) as a function of $Zn^{2+}$ wash-off time from experiments as in (**A**) (n=4–6). Data are plotted as mean ± s.e.m.

The online version of this article includes the following source data for figure 3:

**Source data 1.** Source data for **Figure 3**.

dependence similar to what we observed for mOTOP3 (**Figure 4D**). Activation rates of the more rapidly activating OTOP1 channels were not measurably enhanced by pre-exposure to $Zn^{2+}$ exposure (**Figure 4E**). Thus, potentiation by $Zn^{2+}$ is a feature shared by mOTOP1 and mOTOP3.

## Divalent transition metal ions potentiate and block OTOP3

$Zn^{2+}$ modulates gating of a wide range of ion channels and neurotransmitter receptors, some of which are sensitive to other divalent transition metals interacting with the same residues as $Zn^{2+}$ (**Mathie et al., 2006**; **Shcheglovitov et al., 2012**). For example, the zinc-activated ion channel, a member of the family of Cys-loop receptors, is also activated by copper ($Cu^{2+}$) (**Trattnig et al., 2016**) while the cyclic nucleotide-gated ion channel from rods is potentiated by nickel ($Ni^{2+}$), cadmium ($Cd^{2+}$) and cobalt ($Co^{2+}$), as well as by $Zn^{2+}$, acting through a histidine residue in the mouth of the channel (**Karpen et al., 1993**; **Gordon and Zagotta, 1995a**). Divalent transition metals such as $Co^{2+}$, $Ni^{2+}$, $Cu^{2+}$, and $Cd^{2+}$ are predicted to have distinct preferred coordination geometries in metalloproteins and other proteins to which they bind but are often coordinated by the same acidic and/or polar residues including histidine, glutamic acid, aspartic acid, and cysteine (**Rulíšek and Vondrášek, 1998**). To gain insights into the nature of the $Zn^{2+}$ binding site, we tested whether other transition metals could potentiate mOTOP3 currents. Each metal ion was presented at a concentration of 1 mM for 16 s prior to activation of currents with a pH 5.5 stimulus. Pre-exposure to $Cu^{2+}$ caused a dramatic increase in the magnitudes of the currents, potentiating them by 11.6±0.1-fold (n=10). Pre-exposure to $Cd^{2+}$ and $Ni^{2+}$ had more modest effects, potentiating mOTOP3 currents by 1.7±0.1 (n=10) and 1.6-fold±0.0 (n=8). In contrast, $Co^{2+}$ and iron ($Fe^{2+}$) had little to no effect on the magnitude or kinetics of the currents (**Figure 5**). These results suggest that the binding site occupied by $Zn^{2+}$ may also be shared by other d-block transition metals.

We also tested if the metals that activate also inhibit mOTOP3, like $Zn^{2+}$. Following activation of mOTOP3 by a pH 5.5 stimulus, and in the continued presence of the stimulus, each metal was applied at a concentration of 1 mM. $Cu^{2+}$, $Cd^{2+}$, and $Ni^{2+}$ all inhibited mOTOP3 currents to varying degrees, with $Cu^{2+}$ serving as the most potent inhibitor (**Figure 5—figure supplement 1**). Importantly, $Ni^{2+}$

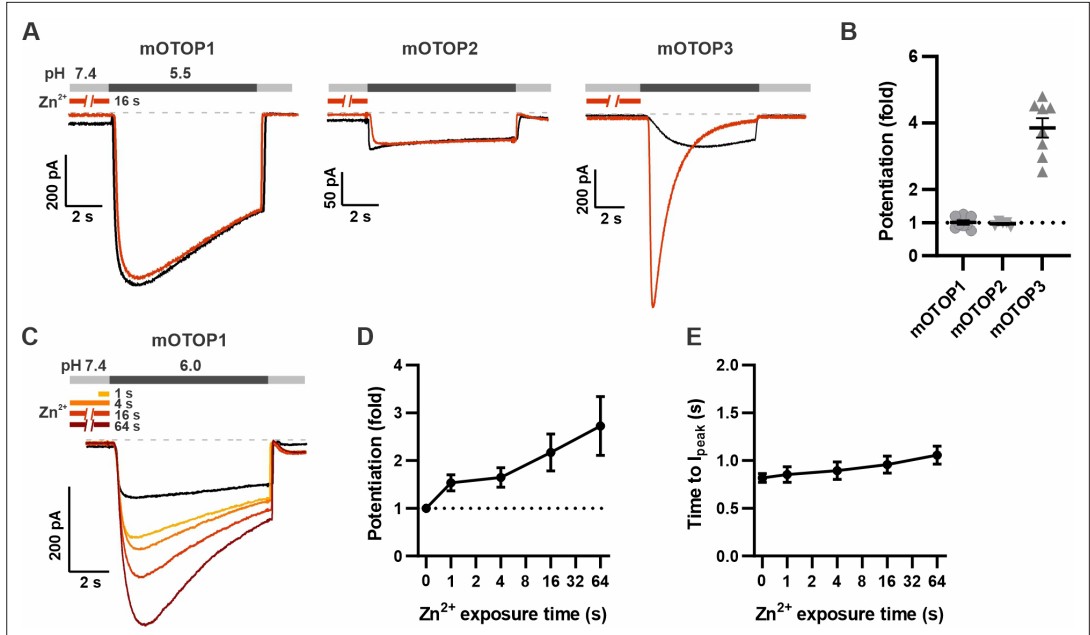

**Figure 4.** mOTOP1 is potentiated by $Zn^{2+}$ when activated by a mild acid stimulus. (**A**) Proton currents recorded from HEK293 cells expressing each of the three mOTOP channels as indicated, in response to pH 5.5 with (red) or without (black) $Zn^{2+}$ pre-exposure. $V_m$ was held at –80 mV. The cells were exposed to 1 mM $Zn^{2+}$ for 16 s prior to pH 5.5 solutions. (**B**) Average and all points data from experiments as in (A) showing the fold potentiation in response to 1 mM $Zn^{2+}$ for 16 s. (**C**) Representative traces showing mOTOP1 currents evoked in response to a pH 6.0 stimulus before and after exposure to $Zn^{2+}$ for varying times as indicated. (**D, E**) The fold potentiation (**C**) and time to $I_{peak}$ (**D**) as a function of $Zn^{2+}$ pre-exposure time from experiments as in (C) (n=5). Data are plotted as mean ± s.e.m.

The online version of this article includes the following source data for figure 4:

**Source data 1.** Source data for *Figure 4*.

inhibited mOTOP3 currents to similar degree as $Zn^{2+}$ but was a less potent activator. We conclude that the inhibitory and activating metal binding sites have different ligand specificity, consistent with the interpretation that they are distinct (see below).

## The tm 11-12 linker is necessary and sufficient to confer sensitivity to $Zn^{2+}$ potentiation

Given the marked difference between mOTOP3 and mOTOP2 in potentiating effects of $Zn^{2+}$, we reasoned that chimeras between the two channels might allow us to identify its structural basis. As $Zn^{2+}$ is likely to bind to an extracellular domain, we tested chimeras in which each of the six external linkers between transmembrane domains were exchanged (*Teng et al., 2022*). A total of twelve chimeras were tested, six in which the backbone was the mOTOP2 channel and six in which the backbone was the mOTOP3 channel. Each chimera was tested for potentiation following pre-exposure to 1 mM $Zn^{2+}$ for 16 s (*Figure 6*).

Strikingly, we found that $Zn^{2+}$ potentiation was eliminated in a chimera containing the mOTOP3 backbone with the tm 11–12 linker from mOTOP2 (O3/O2(L11-12); *Figure 6B, C and E*). Indeed, even a 64 s exposure to 1 mM $Zn^{2+}$ had no effect on the magnitude or activation kinetics of the currents (*Figure 6—figure supplement 1*). Interestingly, the O3/O2(L11-12) chimera is activated at a higher pH and currents are more rapidly activating than currents carried by mOTOP3 (*Figure 6B* and *Teng et al., 2022*). This suggests that the O3/O2(L11-12) channels may be partly locked in a potentiated state. None of the other chimeras with a mOTOP3 backbone showed a loss of potentiation.

Remarkably, simply transplanting the 11–12 linker from mOTOP3 onto mOTOP2 (O2/O3(L11–12)) conferred sensitivity to potentiation by $Zn^{2+}$. Notably, pre-exposure of O2/O3(L11–12) channels to 1 mM $Zn^{2+}$ for 16 s caused a nearly fourfold potentiation of the subsequent currents evoked in response to a pH 5.5. stimulus (*Figure 6A, C and D*). The other five chimeras containing an mOTOP2 backbone remained

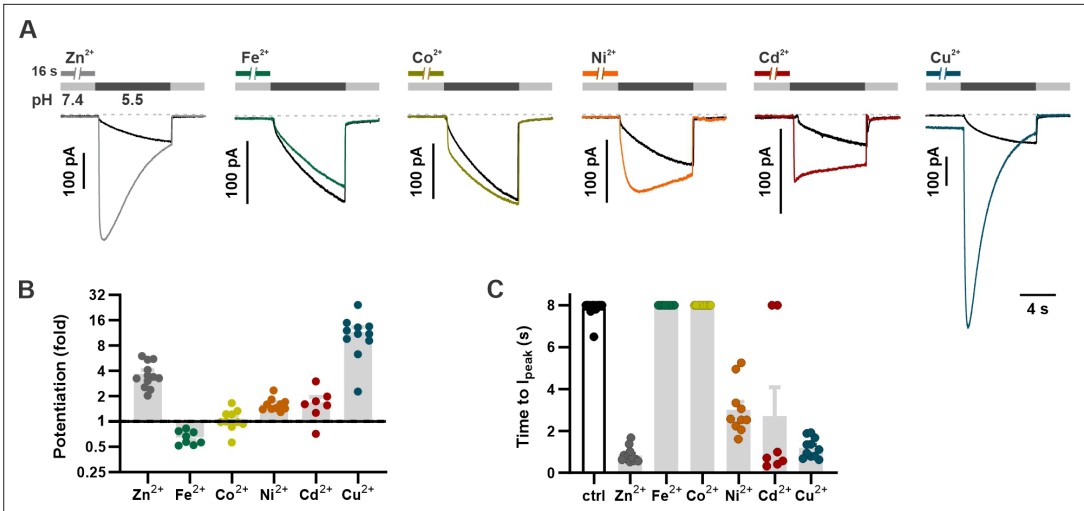

**Figure 5.** Divalent transition metal ions also potentiate mOTOP3. (**A**) Proton currents in response to a pH 5.5 stimulus following exposure (1 mM, 16 s) to various d-block transition metals recorded from HEK293 cells expressing wildtype mOTOP3. $V_m$ was held at –80 mV. Black trace is the control from the same experiment (cell). (**B**) Average (mean ± s.e.m.) and all points data showing the fold potentiation measured from experiments as in (**A**). (**C**) Average (mean ± s.e.m.) and all points data for latency to $I_{peak}$, measured from experiments as in (A). The latency to peak in (C) was scored as 8 s when peak magnitudes were not reached before 8 s.

The online version of this article includes the following source data and figure supplement(s) for figure 5:

**Source data 1.** Source data for *Figure 5*.

**Figure supplement 1.** Divalent transition metals also inhibit mOTOP3.

**Figure supplement 1—source data 1.** Source data for *Figure 5—figure supplement 1*.

resistant to $Zn^{2+}$ potentiation. $Zn^{2+}$ potentiation of O2/O3(L11–12) showed a time dependence similar to that of wildtype mOTOP3 channels (*Figure 6F and G*). mOTOP2 and O2/O3(L11–12), but not mOTOP3, carry outward currents in response to alkaline stimuli (*Teng et al., 2022*). Thus, we wondered if the sensitivity to potentiation by $Zn^{2+}$ would be observed for outward currents. Following exposure to 1 mM $Zn^{2+}$ for 16 s, currents elicited in response to a pH 9.0 stimulus were smaller than in the absence of $Zn^{2+}$ for both WT and O2/O3(L11–12) channels (*Figure 6H*). We conclude that the 11–12 linker contributes to the potentiation of OTOP channels by $Zn^{2+}$ in response to acidic but not alkaline stimuli.

While none of the other chimeras containing an mOTOP3 backbone showed a reduction in $Zn^{2+}$ potentiation, several showed an increase (*Figure 6C and E*). Interestingly, a chimera with a swap of the tm 1–2 linker, O3/O2(L1–2), which was previously observed to be non-conductive in response to changes in extracellular pH (*Teng et al., 2022*), was strongly activated by $Zn^{2+}$. Thus, for this chimeric channel, the increase in fold potentiation mostly reflects the large decrease in the acid-induced currents, rather than an increase in the magnitude of the currents after $Zn^{2+}$ exposure and suggests that the 1–2 linker plays a role in acid activation of mOTOP3.

To determine if potentiating and inhibiting effects of $Zn^{2+}$ were mediated by the same binding site, we next tested whether currents carried by O3/O2(L11–12) and other mOTOP3 chimeras retained sensitivity to inhibition by $Zn^{2+}$ following activation at pH 5.5 (*Figure 7*). All chimeras were inhibited by 1 mM $Zn^{2+}$, including the O3/O2(L11–12) chimera, although the extent of the inhibition, which ranged from ~40% to 80%, varied significantly between some of the chimeras and WT channels (*Figure 7*). Interestingly, potentiation following $Zn^{2+}$ inhibition was absent not just in the O3/O2(L11–12) chimera, as expected, but also in the O3/O2(L3–4) chimera which showed potentiation with the pre-exposure protocol. This suggests that while differences in L3–4 do not account for differences in $Zn^{2+}$ potentiation between mOTOP2 and mOTOP3, residues in L3–4 may nonetheless contribute to activation of the channels.

Together we conclude that the mOTOP3 11–12 linker is both necessary and sufficient to confer sensitivity to potentiation by $Zn^{2+}$. Other parts of the channel may contribute to gating by $Zn^{2+}$ either directly or through allosteric effects.

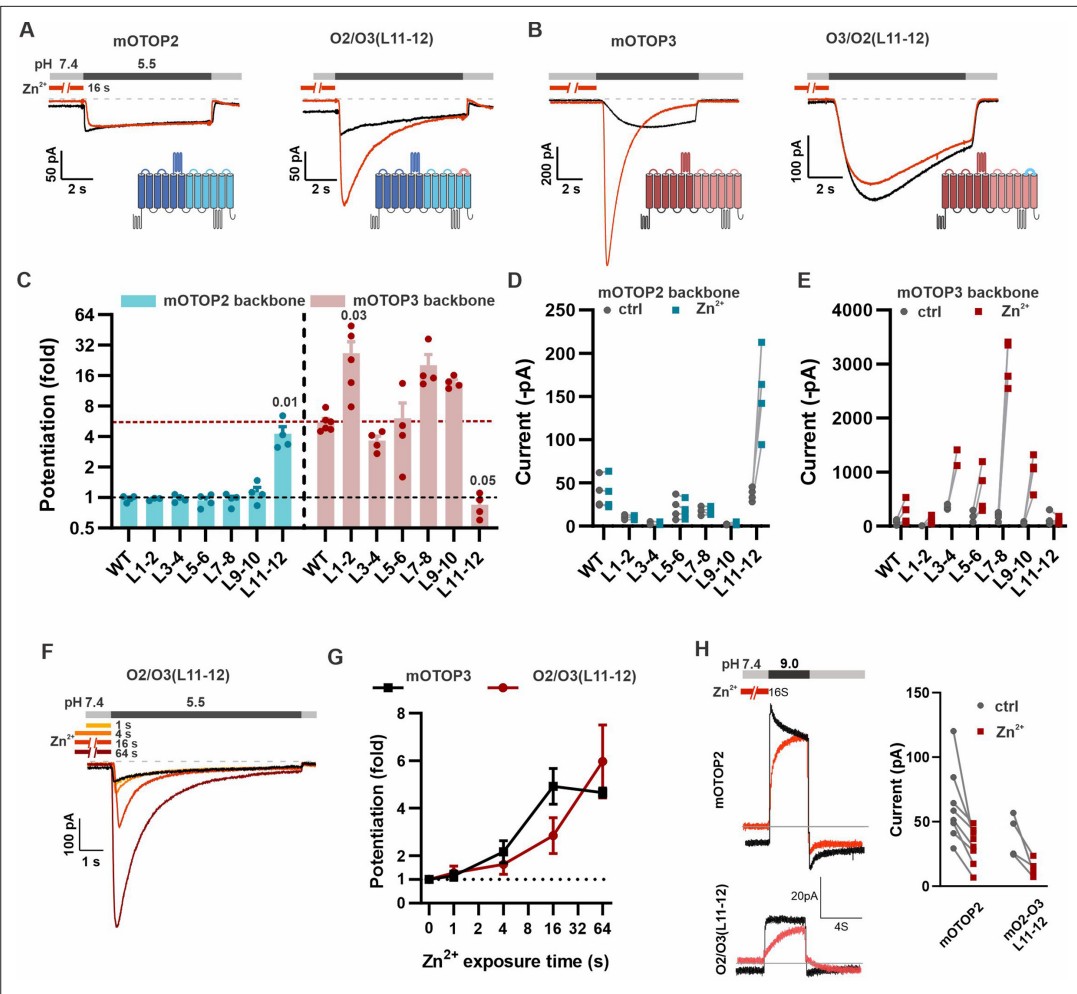

**Figure 6.** The tm 11–12 linker is both necessary and sufficient for Zn²⁺ potentiation. (**A, B**) Proton currents in response to a pH 5.5 stimulus with (red) or without (black) Zn²⁺ pre-exposure (1 mM, 16 s) recorded from HEK293 cells expressing either wildtype (WT) OTOP channels or chimeric channels as indicated. $V_m$ was held at –80 mV. The WT traces are the same set as shown in *Figure 3A*. (**C**) Average data showing the fold potentiation after Zn²⁺ pre-exposure (1 mM, 16 s) measured from experiments as in (**A**) and (**B**). Bars are mean ± s.e.m. mOTOP2 and its chimeras are shown in blue, mOTOP3 and its chimeras are shown in red. Statistical significance determined with an ANOVA using Kruskal-Wallis (non-parametric) statistics. P values are shown where less than 0.05. (**D, E**) Same data as in (**C**) plotted to show current magnitudes before and after Zn²⁺ for WT channels and each of the chimeras. (**F**) Representative traces of O2/O3(L11-12) currents in response to pH 5.5 after pre-exposure to Zn²⁺ for varying times as indicated. (**G**) Average data for experiments as in (**F**) showing the time dependence of the potentiation by Zn²⁺ for the O2/O3(L11-12) chimera as compared with WT (n=4–5). Data of the WT mOTOP3 are the same set as shown in *Figure 2C and D*. (**H**) Left panel: response of mOTOP2 to alkaline stimulus (pH 9.0) with (red) and without (black) Zn²⁺ pre-exposure (1 mM, 16 s). Right panel: magnitude of currents at pH 9 for WT and mutant channels with and without Zn²⁺ pre-exposure. Currents were smaller after Zn²⁺ pre-exposure for both.

The online version of this article includes the following source data and figure supplement(s) for figure 6:

**Source data 1.** Source data for *Figure 6*.

**Figure supplement 1.** O3/O2(L11-12) chimera is completely insensitive to potentiation by Zn²⁺.

**Figure supplement 1—source data 1.** Source data for *Figure 6—figure supplement 1*.

## Contribution of H531 and other residues to Zn²⁺ potentiation

The tm 11–12 linker is relatively short, consisting of sixteen amino acids, of which five are conserved between the three murine OTOP channels (*Figure 8A*). In this region, the residues that could coordinate Zn²⁺ and that vary between mOTOP3 and mOTOP2 are H531, E533, and E535 (mOTOP3

numbering, *Figure 8A*). Within mOTOP3, we mutated each residue to that found in mOTOP2 or to alanine. Strikingly, mutation of H531 to either arginine (found in mOTOP2) or alanine eliminated the ability of $Zn^{2+}$ to potentiate mOTOP3 currents, assessed by measuring either the magnitude of the currents or their activation kinetics (*Figure 8B-D*). However, the converse was not true: introducing histidine at the same position in mOTOP2 (R517H) was not sufficient to produce potentiation by $Zn^{2+}$ under the conditions tested (*Figure 8—figure supplement 1*). mOTOP1 has an arginine at the equivalent residue to H531, but is still potentiated by $Zn^{2+}$, albeit more weakly. Mutation to histidine at this position in OTOP1 (R554H) was sufficient to significantly increase $Zn^{2+}$ potentiation (*Figure 8—figure supplement 1*).

Taken together, we conclude that H531 is a critical element of the $Zn^{2+}$ activating site in mOTOP3. However, the complete binding site is undoubtedly formed by multiple residues that together coordinate $Zn^{2+}$. We, therefore, set out to identify these residues. Mutation of the acidic residues (E533 and E535) within the L11–12 linker had more subtle effects: mutation to residues found in mOTOP2 (H and S, respectively) had no effect on potentiation while mutation of E535 to alanine significantly reduced, but did not eliminate, potentiation (*Figure 8E-H*). Thus, the potentiation of mOTOP3 by $Zn^{2+}$ may involve contributions, either direct or indirect, from both H531 and E535 in the tm 11–12 linker, and the relative contribution of each residue may vary between different channels and under different conditions.

Inspection of the structure of mOTOP3 predicted by AlphaFold (*Jumper et al., 2021*) reveals a possible $Zn^{2+}$ binding site formed by H531, E535, and residues in the linker between tm 5–6, H234, and E238 (*Figure 9B*). Note that a histidine at the position equivalent to 234 (mOTOP3 numbering) is conserved among all murine OTOP channels while glutamic acid is present at a position equivalent to 238 in mOTOP1 (*Figure 9A*). To test the contributions of these residues to $Zn^{2+}$ potentiation, we mutated each alone or in combination to alanine. The single mutations H234A and E238A each showed significantly reduced potentiation (3.5±0.64 and 2.4±0.12-fold potentiation, respectively) as compared with wildtype (10.0±1.0-fold) and the double mutation, H234A/E238A showed a further reduction in potentiation (1.3±0.21) as compared with either of the single mutants (*Figure 9F and G*). Interestingly, all mutants retained some degree of potentiation, and current activation was faster after $Zn^{2+}$ exposure, which was not observed for the mOTOP3 H531A/R single mutation or the mO3_mO2_ L11-12 chimera (*Figure 8D* and *Figure 6—figure supplement 1C*).

The data collectively demonstrate that gating of OTOP channels is regulated by $Zn^{2+}$ acting through residues within the tm 11–12 and tm 5–6 linkers.

## Kinetic model for $Zn^{2+}$ potentiation

These data suggest that $Zn^{2+}$ may act to lock the channel in an open state, possibly by binding more strongly to the same site that is titrated by $H^+$ ions to activate the channel (*Teng et al., 2022*). The data also suggest that a separate site with a faster off-rate mediates inhibition by $Zn^{2+}$. Thus, in the presence of $Zn^{2+}$, the channels may enter a state that is simultaneously activated (open gate) and inhibited. Upon removal of $Zn^{2+}$, the channels may then transit through a fully open state, as the inhibition is relieved faster than the activation. To formally test these predictions, we generated a kinetic model reflecting these properties and asked if it could recapitulate our experimental observations.

We postulated a model comprised of interacting elements that can each transition between two configurations: a pore gate that is either closed or open, a binding site for protons, and two types of binding sites for $Zn^{2+}$ (one activating and one blocking), that are either unoccupied or occupied (*Figure 10A*). For a detailed description of this type of model representation, see *Goldschen-Ohm et al., 2014*. The proton and activating $Zn^{2+}$ sites are energetically coupled to the pore such that binding speeds pore opening and/or slows pore closing. To model the competition of protons and $Zn^{2+}$ for the same activating site(s), we destabilized all states where both the proton and activating $Zn^{2+}$ sites are simultaneously occupied. We modeled the blocking $Zn^{2+}$ site as independent of all other model elements, except that when occupied all channel current is blocked. Simulated currents in response to the same pH and $Zn^{2+}$ protocols used in our experiments qualitatively describe our observed mOTOP3 current responses (*Figure 10B and C*), supporting the plausibility of our proposed mechanism. For example, the model recapitulates the rebound following addition and removal of $Zn^{2+}$ during an acid stimulus. It also recapitulates the speeding up of current activation for a pre-exposure to $Zn^{2+}$ at 0.3 mM, and the increase in current magnitudes with pre-exposure to $Zn^{2+}$ at 1 and 3 mM.

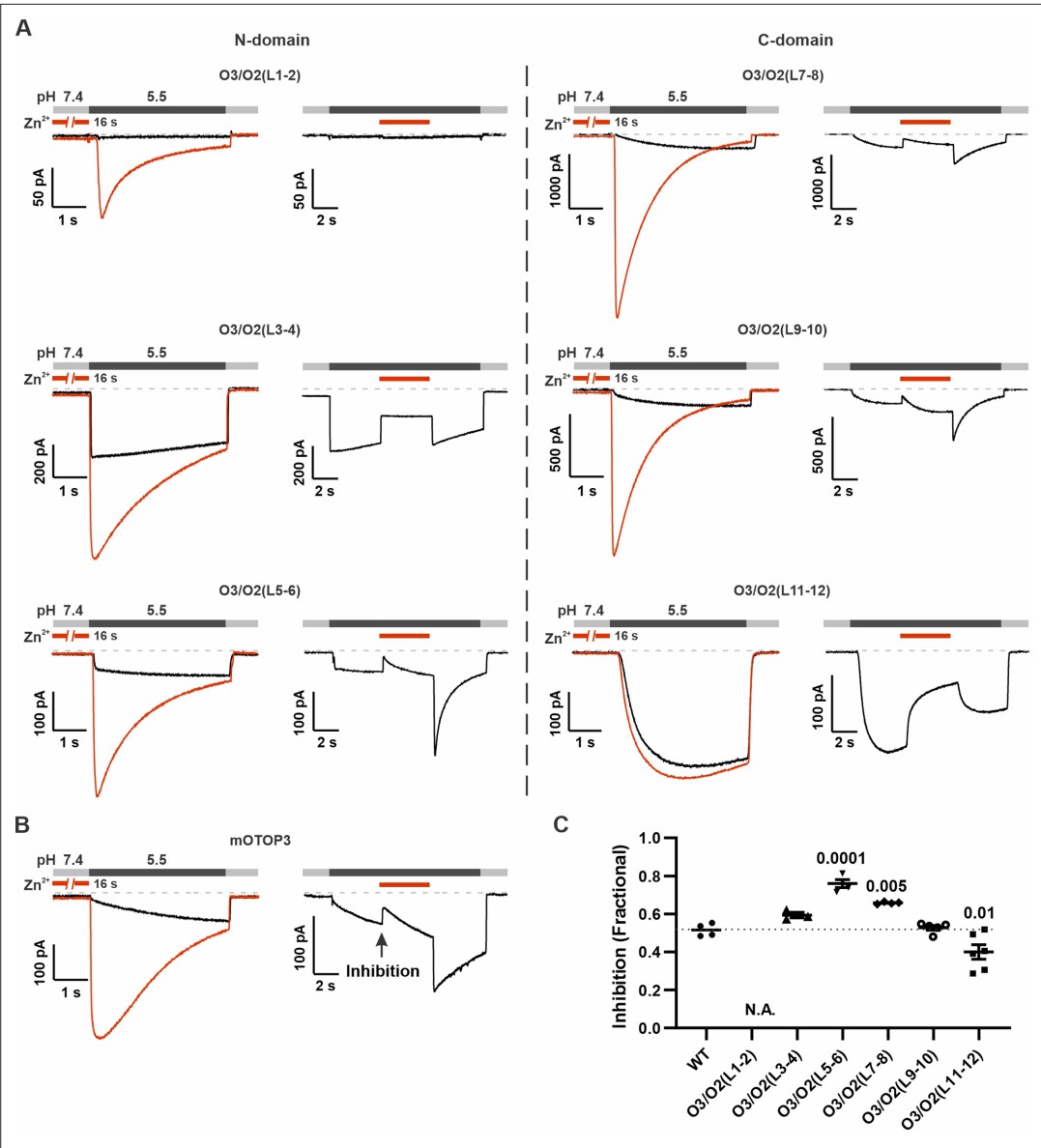

**Figure 7.** Inhibition of mOTOP3 by $Zn^{2+}$ is retained in chimeric channels. (**A**) $Zn^{2+}$ sensitivity of chimeric mOTOP3-mOTOP2 channels as measured with a pre-exposure protocol (left panel in each) or by adding 1 mM $Zn^{2+}$ to the pH 5.5 stimulus (blocking protocol; right panel in each). Chimeras containing mOTOP2 N-domain and C-domain linkers are shown in the left column and right columns, respectively. Data from pre-exposure experiments is also presented in *Figure 6*, and here is shown for comparison to results with the blocking protocol. (B) Representative traces of wildtype mOTOP3 currents in response to the same protocols as in (**A**). The arrow indicates the time point in this trace where inhibition by $Zn^{2+}$ was measured, by comparison with the current magnitude before adding $Zn^{2+}$. (**C**) Average (mean ± s.e.m.) and all points data showing fractional inhibition of currents by 1 mM $Zn^{2+}$ measured from wildtype mOTOP3 and its chimeras. All channels were similarly inhibited by 1 mM $Zn^{2+}$. Significance determined by ANOVA with Dunnett's test corrected for multiple comparisons.

The online version of this article includes the following source data for figure 7:

**Source data 1.** Source data for *Figure 7*.

It also qualitatively recapitulates the decay of the potentiated currents, which in the model is due to unbinding of the $Zn^{2+}$ (see *Figure 10—figure supplements 1–3*). Conceptually, upon occupancy of the activating site by $Zn^{2+}$ the channel will enter a potentiated state, and thereafter upon removal of $Zn^{2+}$ the channel will slowly return to its baseline activity in the absence of $Zn^{2+}$ with a time course that is largely described by the time course of $Zn^{2+}$ unbinding from the activating site (*Figure 10—figure*

*supplement 2*). We note that such a decay could potentially also reflect a desensitization or ion accumulation process which we did not attempt to model.

However, this model also undoubtedly represents an oversimplification of the true number and properties of the states the channel adopts. For example, the model does not recapitulate the observed decay of the currents below baseline when strongly potentiated, which may reflect contributions from an inactivation process or ion accumulation not modeled. Making things more complicated, and interesting, it is also entirely possible that there is more than one permeation pathway and gate (*Chen et al., 2019*; *Saotome et al., 2019*), and that $Zn^{2+}$ opens one of the permeation pathways and not the other. The 11–12 loop sits close to the intrasubunit interface, so that at present it is not possible to predict if it would open a permeation pathway in the N domain, the C domain, or the intrasubunit interface.

## Discussion

The OTOP proton channels were discovered in 2018 when mOTOP1 was cloned from taste tissue as a putative sour receptor (*Tu et al., 2018*). At that time, mOTOP1 was shown to have near-perfect selectivity for protons over other ions, and to generate inward currents as the pH was lowered. Vertebrate OTOP2 and OTOP3, as well as invertebrate channels also carry inward proton currents in response to extracellular acidification, and where tested have been shown to function as proton-selective ion channels (*Tu et al., 2018*; *Chen et al., 2019*) (see also *Teng et al., 2022*). Nearly all descriptions regarding the functional properties of OTOP channels come from the heterologous expression of OTOP channels and save for the description of proton currents now attributed to mOTOP1 channels in taste cells (*Chang et al., 2010*; *Bushman et al., 2015*), all descriptions of OTOP channels postdate the discovery of the genes encoding the proton channels. That is, even by scouring the literature, it is difficult to find a description of a proton current that could be attributed to an OTOP channel in a native cell type. Thus, in contrast to $K^+$ channels where there was a vast literature regarding their functional properties before their cloning, for OTOP channels, this kind of information was not available.

Critically, before this work and that described in *Teng et al., 2022*, it was not known if OTOP channels were gated, and if so by what. Here, we provide evidence that OTOP channels are gated by $Zn^{2+}$ and other transition metals. In the apo, $Zn^{2+}$-free condition, mOTOP3 currents are small and slowly activating. With pre-exposure to 1–3 mM $Zn^{2+}$ for several seconds, the currents increase in magnitude by up to 10-fold. This can only be explained if $Zn^{2+}$ either increases the probability that the channels open or produces a change in their conductance – gating them. We also find that pre-exposure to copper ($Cu^{2+}$) strongly potentiates OTOP3 currents. Together with other recent evidence for pH-dependent gating (*Teng et al., 2022*), we can now say with certainty that some OTOP channels are gated.

### Structural considerations

The structures of two of the three OTOP channels have been reported at near-atomic resolution ([zebrafish OTOP1, chicken OTOP3, and *Xenopus* OTOP3; *Chen et al., 2019*; *Saotome et al., 2019*)]. In addition, predictions are available for structures of mouse and human OTOP channels that appear to be reliable (*Jumper et al., 2021*; *Teng et al., 2022*; *Varadi et al., 2022*). All the structures to date share common features: The protomers assemble as dimers, and each protomer shows a twofold symmetry, leading to an overall pseudo-tetrameric stoichiometry. The four pseudo-subunits come together to form a central cavity that is filled with lipids and cannot support ion permeation. Instead, ions may permeate through the barrel-like structures formed from transmembrane domains 2–6 (N domain) and 8–12 (C domain) or at the intrasubunit interface (formed mostly by tm 6 and 12). From these static structures, it is not possible to tell if the channels are gated, and if so, what state they are

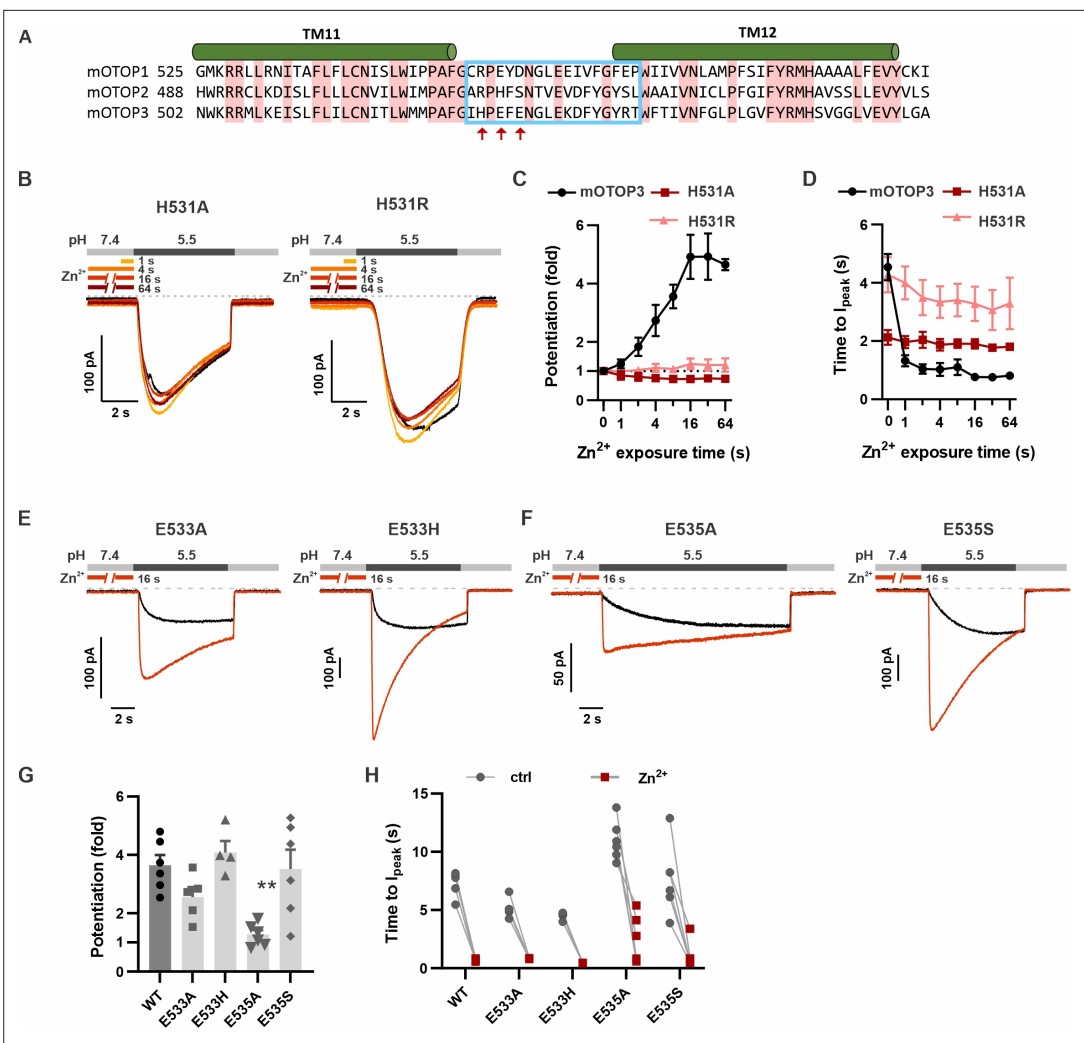

**Figure 8.** H531 in mOTOP3 L11–12 is essential for $Zn^{2+}$ potentiation. (**A**) Sequence alignment of three mOTOP channels. The residues that were exchanged between mOTOP2 and mOTOP3 in the L11–12 chimeras is indicated with a blue box. Residues that differed between the two channels and that were tested are indicated by red arrows. (**B**) Representative traces of mOTOP3_H531A and H531R currents in response to pH 5.5 after pre-exposure to $Zn^{2+}$ for varying times as indicated. (**C, D**) Average data for fold potentiation (**C**) and latency to $I_{peak}$ (**D**) measured from experiments as in (**B**), plotted as a function of pre-exposure time to $Zn^{2+}$ (n=3–7). Data from wildtype (WT) mOTOP3 are the same set as shown in *Figure 2C and D*. (**E, F**) Responses of mOTOP3 mutants as indicated in response to pH 5.5 with (red) or without (black) $Zn^{2+}$ pre-exposure (1 mM, 16 s). $V_m$ was held at –80 mV. (**G**) Average data for fold potentiation measured from experiments as in (**E and F**). Statistical significance compared with WT determined using the Kruskal-Wallis (non-parametric) test corrected for multiple comparison. (**H**) Latency to peak currents of WT mOTOP3 or mutant currents, measured from experiments as in (**E and F**). Each set of points represents a separate cell.

The online version of this article includes the following source data and figure supplement(s) for figure 8:

**Source data 1.** Source data for *Figure 8*.

**Figure supplement 1.** Introduction of histidine into mOTOP1 partially confers sensitivity to potentiation by $Zn^{2+}$.

**Figure supplement 1—source data 1.** Source data for *Figure 8—figure supplement 1*.

in (open or closed). Based on the pH sensitivity of the channels (*Teng et al., 2022*), we presume they are closed.

We have focused on mOTOP2 and mOTOP3, which show the most divergent functional properties. In addition to differences in the effects of $Zn^{2+}$ described here, we have also described differences in pH sensitivity of the two channels (*Teng et al., 2022*). mOTOP3 is gated by protons, and thus

conducts currents only in response to extracellular acidification (<pH 5.5), while OTOP2 is constitutively open and conducts currents over a large pH range, including outward currents when the extracellular solution is alkalinized. The strikingly different functional properties of the two channels, but otherwise overall similar architecture, provided us the opportunity to identify motifs involved in gating using a chimeric approach.

Remarkably, we found that a short stretch of amino acids linking transmembrane domains 11–12 was necessary for $Zn^{2+}$ potentiation of mOTOP3 and sufficient to confer $Zn^{2+}$-sensitive gating on mOTOP2. Within that stretch, we identified one amino acid, H531, mutations of which (to R or A)

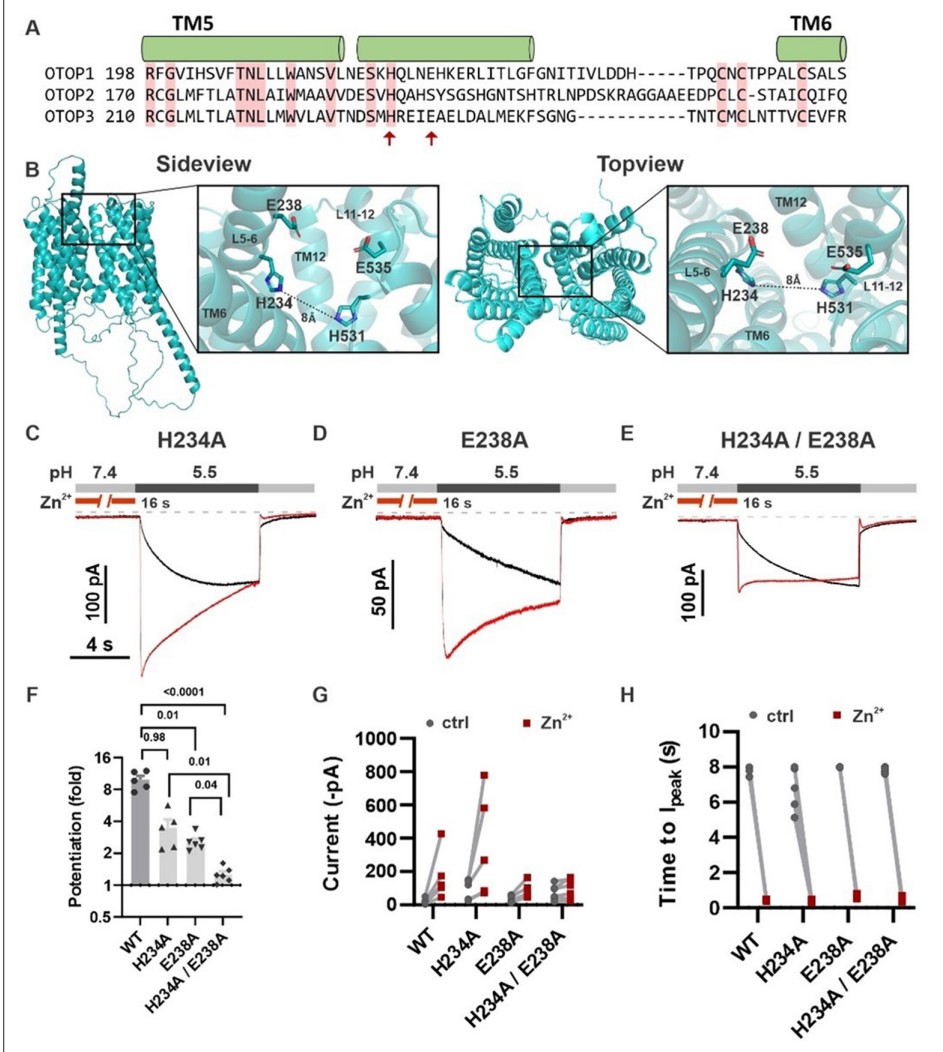

**Figure 9.** H234 and E238 in transmembrane domain 5 contribute to $Zn^{2+}$ potentiation of mOTOP3. (**A**) Sequence alignment of the three mOTOP channels. Alpha helices shown above the sequence are based on the AlphaFold prediction of the structure of mOTOP3. Red arrows indicate residues neutralized to alanine in subsequent experiments. (**B**) Images generated using the AlphaFold predicted structure of mOTOP3. Left panel shows a sideview and right panel shows a topview of mOTOP3. Each zoom-in highlights the predicted $Zn^{2+}$ potentiation binding site. (**C,D,E**) Representative traces of mOTOP3 H234A, E238A, and a double mutation of H234A/E238A in response to lowering the extracellular pH to 5.5 after (red) and without (black) pre-exposure to $Zn^{2+}$ (1 mM, 16 s). $V_m$ was held at –80 mV. (**F**) Average (mean ± s.e.m.) and all points data showing the fold potentiation measured from experiments as in (**C,D,E**). Statistical significance determined using an ANOVA with the Kruskal-Wallis (nonparametric) test. (**G,H**) Same data as in (**F**) plotted to show current magnitudes before and after $Zn^{2+}$ (**G**) and the time to $I_{peak}$ with and without pre-exposure to $Zn^{2+}$ (**H**) from experiments in (**C,D,E**).

The online version of this article includes the following source data for figure 9:

**Source data 1.** Source data for *Figure 9*.

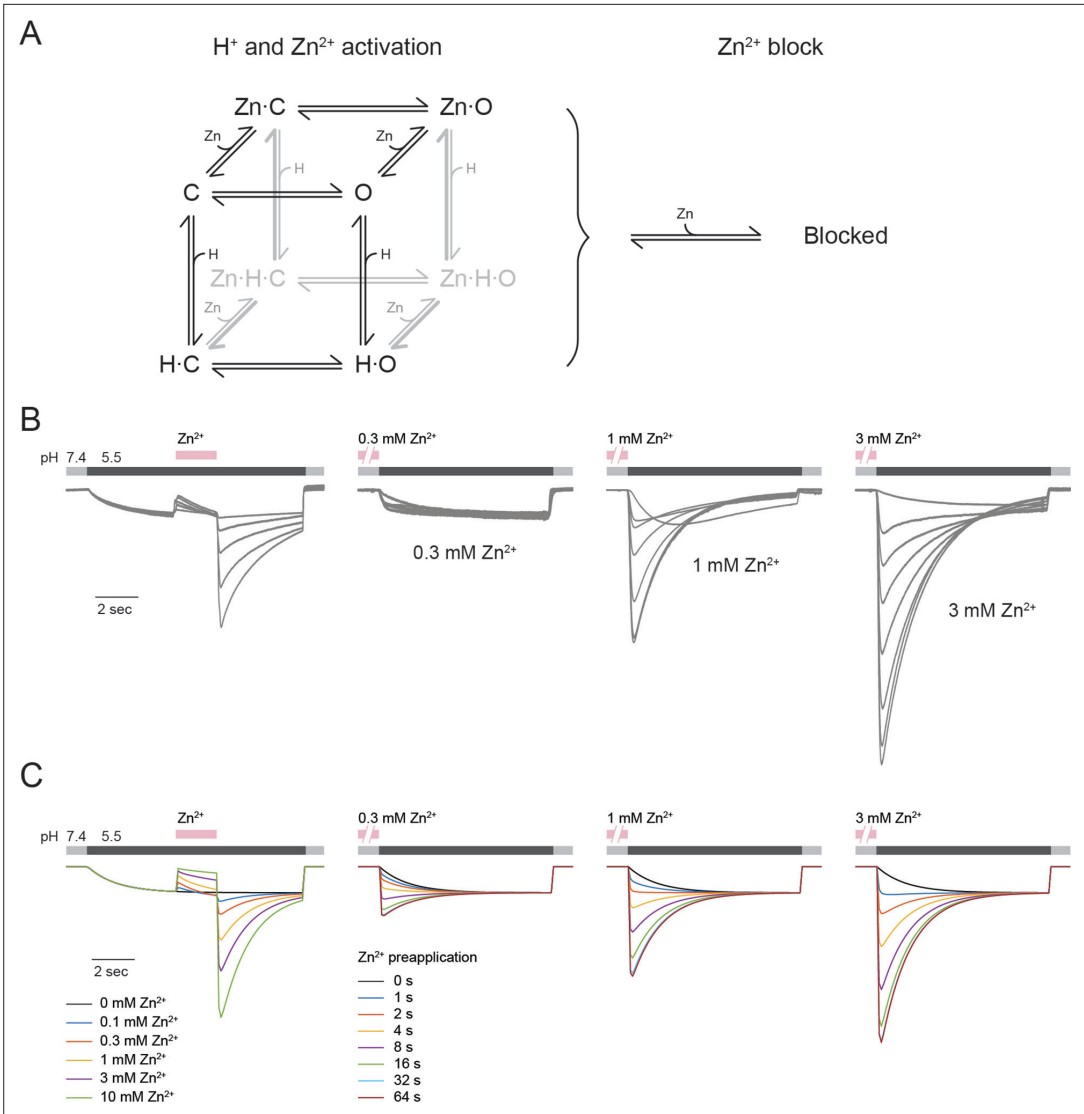

**Figure 10.** mOTOP3 kinetic model for Zn$^{2+}$ potentiation and block. (**A**) Kinetic model for activation of mOTOP3 by H$^+$ and Zn$^{2+}$. The channel moves from a closed state (**C**) upon binding H$^+$ or Zn$^{2+}$ to an open state (Zn-O, H–O) in which it permeates protons. The doubly bound Zn-H-O and Zn-H-C states are disfavored energetically. A separate site binds Zn$^{2+}$ and inhibits channel permeation, independently of the gating state. See methods for a more detailed description of the model. (**B**) Example mOTOP3 current recordings from *Figure 1C* and *Figure 2B*. Each set of recordings was from a different cell. (**C**) Simulated currents for the model under the same protocols as in (B). Model parameters were adjusted manually and were the same for all traces. The model replicates the rebound seen upon addition of Zn$^{2+}$ at pH 5.5, and the potentiation seen with pre-exposure to Zn$^{2+}$ at pH 7.4.

The online version of this article includes the following figure supplement(s) for figure 10:

**Figure supplement 1.** Simulation time course for individual gating and binding domains in response to a pulse of Zn$^{2+}$ during an acid stimulus.

**Figure supplement 2.** Simulated decay of current potentiation largely reflects Zn$^{2+}$ unbinding from the activating site.

**Figure supplement 3.** Simulation time course for individual gating and binding domains in response to an acid stimulus following pre-exposure to Zn$^{2+}$.

completely abolished $Zn^{2+}$-sensitive gating in mOTOP3. As histidine is well documented to form part of the $Zn^{2+}$ binding site in other $Zn^{2+}$-sensitive proteins (**Vallee and Auld, 1990**), including the voltage-gated proton channel HV1 (**Ramsey et al., 2006**), H531 likely contributes to coordinating $Zn^{2+}$ in mOTOP3. However, $Zn^{2+}$ is typically coordinated by sidechains of four or more residues, which in addition to the imidazole rings of histidine, includes sulfhydryl groups of cysteine, carboxyl group of acidic residues (aspartic and glutamic acid). Moreover, water can also participate in coordinating $Zn^{2+}$ (**Vallee and Auld, 1990**). In addition to H531, we have identified three other residues, H234, E238, and E535, that when mutated to alanine lead to reduced efficacy of $Zn^{2+}$ to potentiate mOTOP3. Based on the functional data provided in this paper and an AlphaFold structure of mOTOP3, we tentatively predict these four residues form the $Zn^{2+}$ activating binding site. Interestingly, alkali activation of mOTOP1 was recently shown to require a basic residue at a position equivalent to H531 in mOTOP3 and in the tm 11–12 linker at a position close to H234 (**Tian et al., 2023**). It is tempting to speculate that $Zn^{2+}$ binding could bring together residues in the tm 5–6 linker and tm 11–12 linker that are at a distance in the closed state (8 Å; **Figure 9B**) to stabilize an open state, as shown for other ion channels (**Gordon and Zagotta, 1995b**; **Zhang et al., 2009**). A similar mechanism could open mOTOP1 channels in response to alkaline pH.

## Role of zinc in health and disease

Zinc is required for the functioning of a wide range of enzymes and proteins in the body and supplementary zinc has both beneficial and detrimental health effects. Given that it is mostly not known which cells express OTOP channels, or what functions they play in those cells, it is hard to predict the functional role of $Zn^{2+}$-sensitive gating in physiology. The dual blocking and potentiating effect of $Zn^{2+}$ make it likely that the enhanced activity would be mostly evident under conditions where $Zn^{2+}$ concentrations dropped rapidly or that favored binding of $Zn^{2+}$ to its activating site over its blocking site. These studies also raise the interesting prospect that other molecules could be found that function as positive allosteric modulators of OTOP channels, with selectivity for the activating site over the blocking site, and that could ameliorate effects of reductions in OTOP channel function that lead to vestibular or other disorders.

# Materials and methods

## Key resources table

| Reagent type (species) or resource | Designation | Source or reference | Identifiers | Additional information |
|---|---|---|---|---|
| Gene (*Mus musculus*) | *Otop1, Otop2,* and *Otop3* | **Tu et al., 2018**. PMID:29371428 | | |
| Cell line (*Homo sapiens*) | HEK293 | ATCC | CRL-1573 | |
| Cell line (*Homo sapiens*) | PAC-KO HEK293 cells | **Yang et al., 2019**. PMID:31023925 | | |
| Recombinant DNA reagent | *Otop1, Otop2* and *Otop3* in pcDNA3.1 | **Tu et al., 2018**. PMID:29371428 | | |
| Recombinant DNA reagent | *Otop1, Otop2* and *Otop3* – GFP | **Saotome et al., 2019**. PMID:31160780 | | |
| Recombinant DNA reagent | mO2_O3 loop swap mutations | This paper | | cDNAs encode chimeric channels (see Materials and methods and **Figure 6—figure supplement 1**). Available upon request |
| Recombinant DNA reagent | mO3_O2 loop swap mutations | This paper | | cDNAs encode chimeric channels (see Materials and methods and **Figure 6—figure supplement 1**). Available upon request |
| Recombinant DNA reagent | pHluorin in pcDNA3 | Miesenbock, et al., 1998. PMID:9671304 | | |
| Chemical compound, drug | CHES | Sigma | C2885 | |
| Chemical compound, drug | PIPES | Sigma | P6757 | |
| Chemical compound, drug | Homopiperazine-1,4-bis(2-ethanesulfonic acid) | Sigma | 53588 | |

*Continued on next page*

*Continued*

| Reagent type (species) or resource | Designation | Source or reference | Identifiers | Additional information |
|---|---|---|---|---|
| Software, algorithm | GraphPad Prism 8 and 9 | GraphPad | RRID:SCR_002798 | |
| Software, algorithm | pClamp and clampfit | Molecular Devices | RRID:SCR_011323 | |
| Software, algorithm | Origin | OriginLab corporation | RRID:SCR_002815 | |
| Software, algorithm | CorelDraw | Corel | RRID:SCR_014235 | |
| Software, algorithm | SimplePCI | HCImage | https://hcimage.com/simple-pci-legacy/ | |

## Clones, cell lines, and transfection

Mouse OTOP1, OTOP2, and OTOP3 cDNAs were in a pcDNA3.1 vector with an N-terminal fusion tag consisting of an octahistidine tag followed by eGFP, a Gly-Thr-Gly-Thr linker, and 3C protease cleavage site (LEVLFQGP) were as previously described (*Saotome et al., 2019*). All chimeras and mutations were generated using In-Fusion Cloning (Takara Bio) and were confirmed by Sanger sequencing (Genewiz) (see *Teng et al., 2022*).

HEK293 cells were purchased from ATCC (CRL-1573). The cells were cultured in a humidified incubator at 37°C in 5% $CO_2$ and 95% $O_2$. The high glucose DMEM media (Thermo Fisher) is implemented with 10% fetal bovine serum (Life Technology) and 1% penicillin-streptomycin antibiotics. Cells were passaged every 3–4 days. Cells were tested and found free of mycoplasma using a PCR detection kit (Sigma-Aldrich, USA).

Cells used for patch-clamp recordings were transfected in 35 mm Petri dishes, with 600–1000 ng DNA and 2 µL TransIT-LT1 transfection reagents (Mirus Bio Corporation) following the manufacturer's protocol. After 24 hr, the cells were lifted using trypsin-EDTA and plated into a recording chamber.

## Patch-clamp electrophysiology

Whole-cell patch-clamp recording was performed as previously described (*Teng et al., 2019*). Briefly, recordings were obtained with an Axonpatch 200B amplifier, digitized with a Digidata 1322a 16-bit data acquisition system, acquired with pClamp 8.2, and analyzed with Clampfit 9 or 10 (Molecular Devices). Records were sampled at 5 kHz and filtered at 1 kHz. Patch pipettes with a resistance of 2–4 MΩ were fabricated from borosilicate glass (Sutter Instrument). Solution exchange was achieved with a fast-step perfusion system (Warner Instrument, SF-77B) custom modified to hold seven microcapillary tubes in a linear array. Cells were treated with trypsin-EGTA and plated into the recording chamber immediately before each experiment. After a gigaohm seal was formed and whole-cell recording was achieved, the cell was lifted and moved in front of the microcapillary tubes. The membrane potentials were held at −80 mV unless otherwise indicated.

## Patch-clamp electrophysiology solutions

Tyrode's solution contained 145 mM NaCl, 5 mM KCl, 1 mM $MgCl_2$, 2 mM $CaCl_2$, 20 mM dextrose, 10 mM HEPES (pH adjusted to 7.4 with NaOH). Standard pipette solution contained 120 mM Cs-aspartate, 15 mM CsCl, 2 mM Mg-ATP, 5 mM EGTA, 2.4 mM $CaCl_2$ (100 nM free $Ca^{2+}$), and 10 mM HEPES (pH adjusted to 7.3 with CsOH; 290 mOsm). Standard $Na^+$-free extracellular solutions contained 160 mM NMDG, 2 mM $CaCl_2$, and 10 mM buffer (HEPES for pH 7.4, MES for pH 6.0 and 5.5), and pH was adjusted with HCl to 7.4.

$ZnCl_2$ was directly introduced into the $Na^+$-free solutions where the final concentration was less than 3 mM, which caused a change in osmolarity of <10 mOsm. The pH was re-adjusted with NMDG-OH or HCl as needed. For the experiment in *Figure 1C*, the concentration of NMDG in the 10 mM $Zn^{2+}$ solution was reduced to maintain an ~300 mOSM osmolarity.

For experiments in *Figure 5* and *Figure 5—figure supplement 1*, $FeCl_2$, $CoCl_2$, $NiCl_2$, $CuCl_2$, and $CdCl_2$ were directly introduced into the $Na^+$-free solutions at a concentration of 1 mM immediately before the experiment was performed. The pH was re-adjusted with NMDG-OH or HCl as needed.

## Quantification and statistical analysis

All data are presented as mean ± SEM if not otherwise noted. Statistical analysis was performed using GraphPad Prism 9 (GraphPad Software Inc). The sample sizes of 3–10 independent recordings from individual cells per data point are similar to those in the literature for similar studies. For comparison of fold potentiation, an ANOVA was used with the Kruskal-Wallis (non-parametric) test (GraphPad Prism 9). All replicates are biological replicates. Number of replicates are indicated in the figure or figure legend. They represent recordings from different cells transfected with the same plasmid DNA. Data were excluded if a gigaohm seal was not established or maintained, as indicated by an inward current of >80 pA (–80 mV) in the presence of the OTOP channel blocker $Zn^{2+}$ applied at pH 7.4. Data sets that represent time series were excluded if four or fewer time points were collected out of a possible eight time points, due to seal instability.

Representative electrophysiology traces shown in the figures were acquired with pClamp, and in some cases, the data was decimated by 10-fold before exporting into graphic programs, Origin (Microcal) and Coreldraw (Corel).

## Kinetic modeling

A kinetic model was generated based on interacting binary elements as described in *Goldschen-Ohm et al., 2014*. Briefly, each element can adopt one of two possible configurations (closed or open for the pore, unbound or bound for each binding site) with intrinsic rate constants for transitioning between them. The elements are energetically coupled such that binding at a particular site either promotes or inhibits pore opening. To simulate competition between $H^+$ and $Zn^{2+}$ for the activating site(s), occupation of one site energetically destabilizes the other.

Model parameters were manually adjusted to qualitatively recapitulate the experimental observations. Rate constants for pore opening/closing ($k_o/k_c$), proton binding/unbinding ($k_{onH}/k_{offH}$), $Zn^{2+}$ binding/unbinding to activating sites ($k_{onZnA}/k_{offZnA}$) and blocking site ($k_{onZnB}/k_{offZnB}$) are ($s^{-1}$ or $M^{-1}s^{-1}$ for binding rates): $k_o = 0.02$, $k_c = 25$, $k_{onH} = 5 \times 10^4$, $k_{offH} = 1$, $k_{onZnA} = 35$, $k_{offZnA} = 1$, $k_{onZnB} = 5 \times 10^4$, and $k_{offZnB} = 20$. In *Figure 10* these rates reflect the transition pairs from the unbound closed state C to either the unbound open state O ($k_o$, $k_c$) or each closed state having only a single binding site occupied (H·C, Zn·C, Blocked·C) ($k_{onH}$, $k_{offH}$, $k_{onZnA}$, $k_{offZnA}$, $k_{onZnB}$, $k_{offZnB}$). To describe how binding to a site influences pore opening and/or binding to other sites, and vice versa, we define state-dependent interaction energies between the pore gate and binding site domains. These energies modulate the rate constants as defined in *Goldschen-Ohm et al., 2014*. The interaction between the proton site and pore gate is such that proton binding reduces the energy barrier for pore opening by –4 kcal/mol without affecting pore closure. For example, this means that the transition rate from H·C to H·O will be exp(–(–4 kcal/mol)/RT)~850 times faster than that from C to O, where RT is the product of the molar gas constant R and temperature T and has value 0.593 kcal/mol at a room temperature of 298 K. Reciprocally, pore opening reduces the energy barrier for proton binding by –1 kcal/mol. These energetic effects imply that pore opening must also increase the barrier for proton unbinding by 3 kcal/mol (assuming transition rates are described by a single transition state energy barrier). The interaction between the activating $Zn^{2+}$ site and pore gate is such that $Zn^{2+}$ binding both reduces the energy barrier for pore opening by –3 kcal/mol and increases the barrier to pore closure by 3 kcal/mol, thereby increasing opening frequency and stabilizing the open pore. Reciprocally, pore opening reduces the barrier for $Zn^{2+}$ binding to the activating site by –5 kcal/mol. These energetic effects imply that pore opening must also increase the barrier for $Zn^{2+}$ unbinding from the activating site by 1 kcal/mol, thereby stabilizing bound $Zn^{2+}$ at the activating site. To simulate competition between protons and $Zn^{2+}$ for the activating site, states with both proton and activating $Zn^{2+}$ sites occupied were destabilized by 6 kcal/mol, the value for which was simply chosen to be a

somewhat large destabilizing energy. The $Zn^{2+}$ blocking site was assumed to be independent of all other elements, but when occupied blocks all channel current. This model includes 14 free parameters: eight rate constants describing the opening/closing of the pore gate and binding unbinding at the proton and $Zn^{2+}$ activating and blocking sites, and six state-dependent interaction energies defining how activation/occupation of the above elements influences the rates of the other elements. However, the model is relatively insensitive to some of the parameters (within reason). For example, binding/unbinding of $Zn^{2+}$ from the blocking site is very fast in comparison to the macroscopic current kinetics of proton activation and decay of $Zn^{2+}$ potentiation. Thus, any set of blocking kinetics that give a relatively rapid onset and offset of the block will likely perform similarly in the model.

For comparison with model simulations, the entire set of currents across all cells was uniformly scaled under the assumption that the maximal response following preapplication of 3 mM $Zn^{2+}$ is reflective of channels with an open probability of ~0.9, a measure that has not been experimentally verified. It is likely that the same model structure will similarly describe responses with lower maximal open probabilities, albeit with slightly different parameter values. Given these uncertainties, as well as the relatively large number of free parameters (11 parameters defining pore gating, binding to activating sites, and their interactions are most relevant), we did not attempt to optimize the model parameters, but instead manually identified parameters that qualitatively recapitulate the data. The time-dependent probability in each state was simulated as described by *Colquhoun and Hawkes, 1995*, after first generating the matrix of transition rates between states from the model's binary elements and interactions (*Goldschen-Ohm et al., 2014*). Currents were computed from the simulated open probability assuming a conductance of 1 pS and a driving force of –80 mV in pH 5.5 and 0 in pH 7.4 as estimated from ramp experiments (*Teng et al., 2022*). The choice of conductance is arbitrary given that we do not know how many channels are in each recording, and thus only contributes to the overall scaling of the current.

## Material availability

All materials generated during this study including mutant channels are available upon request.

## Data availability

All data generated or analyzed during this study are included in the manuscript and supporting files. The source code for simulations is provided in supplementary information.

## Acknowledgements

We thank Jackson Walker and Anne Tran for expert technical support, Carly Hamel and Roei Zakut, for help in generating mutant channels, all members of the Liman lab for helpful discussions, and all the reviewers of the manuscript for their constructive comments, including the suggestion to include an assessment of contributions of residues in the tm 5–6 linker. This research was supported by NIH grants R01GM131234 to ERL and R01GM148591 to MG-O.

## Additional information

### Competing interests

Marcel P Goldschen-Ohm: Reviewing editor, *eLife*. The other authors declare that no competing interests exist.

## Funding

| Funder | Grant reference number | Author |
|---|---|---|
| National Institutes of Health | R01GM131234 | Emily R Liman |
| National Institutes of Health | R01GM148591 | Marcel P Goldschen-Ohm |

The funders had no role in study design, data collection and interpretation, or the decision to submit the work for publication.

## Author contributions

Bochuan Teng, Conceptualization, Data curation, Formal analysis, Investigation, Visualization, Methodology, Writing – original draft, Writing – review and editing; Joshua P Kaplan, Conceptualization, Data curation, Formal analysis, Investigation, Writing – original draft, Writing – review and editing; Ziyu Liang, Formal analysis, Investigation; Kevin Saejin Chyung, Investigation; Marcel P Goldschen-Ohm, Conceptualization, Software, Formal analysis, Funding acquisition, Investigation, Writing – original draft, Writing – review and editing; Emily R Liman, Conceptualization, Resources, Formal analysis, Supervision, Funding acquisition, Visualization, Methodology, Writing – original draft, Project administration, Writing – review and editing

## Author ORCIDs

Bochuan Teng ⓘ http://orcid.org/0000-0001-7918-484X
Joshua P Kaplan ⓘ http://orcid.org/0000-0001-5064-2130
Marcel P Goldschen-Ohm ⓘ http://orcid.org/0000-0003-1466-9808
Emily R Liman ⓘ http://orcid.org/0000-0003-4765-5496

## Decision letter and Author response

Decision letter https://doi.org/10.7554/eLife.85317.sa1
Author response https://doi.org/10.7554/eLife.85317.sa2

# Additional files

## Supplementary files

• Transparent reporting form
• Source code 1. MATLAB scripts for running the kinetic model simulations shown in *Figure 10*.

## Data availability

All data generated or analysed during this study are included in the manuscript and supporting files.

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
