## [Editor Report]

This valuable study discovers that zinc ions can activate some OTOP proton channels, identifying a pharmacological tool for research, and further establishing that OTOP channels gate. The data presented provides convincing support for conclusions. This work is expected to be of great interest to physiologists studying OTOP channels and other proton-permeation pathways.

---

## [Decision Letter]

**Decision letter after peer review:**

Thank you for submitting your article "Zn^2+^ potentiation of OTOP proton channels identifies structural elements of the gating apparatus" for consideration by *eLife*. Your article has been reviewed by 3 peer reviewers, including Leon D Islas as Reviewing Editor and Reviewer #1, and the evaluation has been overseen by Kenton Swartz as the Senior Editor. The following individual involved in the review of your submission has agreed to reveal their identity: Jon T Sack (Reviewer #2).

Essential revisions:

After consultation, the reviewers concur on these essential revisions to the manuscript.

1) In the explanation of the model predictions, it is not clear how the allosteric model can predict the desensitization of currents. This part needs a more detailed explanation. Also, it was difficult to tell if the description in methods of the kinetic model fully describes the model. There appear to be 24 different rate constants. Perhaps labeling a table listing all the free parameters using a nomenclature clearly addressable to Figure 9A would help readers understand how the model works.

2) Given the effect of H531 in OTOP3 it is reasonable to wonder if the histidine is sufficient to recapitulate the O3/O2tm11-12 phenotype or at least part of it. The same position should be replaced with histidine in OTOP2 and OTOP1, which have arginines. Related to this point: can the authors provide an explanation for the OTOP1 potentiation given that this isoform does not have the corresponding H531 residue?

3) Please comment on to what extent the desensitization of the proton currents could be a reflection of proton depletion given the large amplitude of the proton currents.

Also please attend to the specific comments of reviewers below. Only the experiments in section 2 above are viewed as essential.

*Reviewer #1 (Recommendations for the authors):*

The authors of this manuscript have discovered that the recently characterized proton (H^+^) gated, proton permeable channel OTOP3 is strongly potentiated by the extracellular application of zinc (Zn2+) ions, while the related isoform OTOP2 is not. After characterizing the main features of the potentiation effects, the authors show, through the construction of chimeric channels, that one of the extracellular linkers, the one between transmembrane domains 11-12 is capable of encoding the potentiation phenotype. Further experiments demonstrate that a histidine residue at position 531 in the 11-12 linker is mainly responsible for Zn2+ potentiation of OTOP3 channels.

In an attempt to provide a quantitative explanation for their findings, the authors produce a transition-state gating model that is capable of explaining most of the experimental findings by postulating that gating is influenced by the interaction of two separate binding sites for H^+^ and Zn2+.

The experiments and analyses performed in this manuscript are of very high quality and the interpretation of the data and conclusions are supported by the data.

This is a contribution that should help understand the emerging physiological roles of this recent addition to the family of proton permeable channels.

Specific comments:

1) In several places in the manuscript, the authors mention that Zn potentiation indicates that the channels are gated, however, isn't activation by H an indication that the channels are are gated? Perhaps this statement should be explained in a more nuanced manner.

2) The solutions employed have a relatively low concentration of buffer, thus the proton currents will incur in proton depletion artifacts, especially for large amplitude currents. Please comment on to what extent the desensitization of the proton currents could be a reflection of proton depletion.

3) The data in figure 3A shows the time course of the wash-off of the Zn effect. What process do the authors think is reflected here? A slow k_off_ for Zn unbinding or a slow rate constant for recovery from the potentiated state?

4) In the methods section, please explain in more detail the procedure to perfuse cells in whole-cell recordings. Were the cells lifted and brought to the perfusion tubes? Or were the cells attached and the surrounding solution changed? If the latter, how complete is the solution exchange to be expected?

5) In the explanation of the model predictions, it is not clear to me how the allosteric model can predict the desensitization of currents. This part needs a more detailed explanation.

*Reviewer #2 (Recommendations for the authors):*

It was curious that it didn't seem like potentiation could occur in the presence of Zn. Despite the identification of an important His residue and the different concentration response of inhibition vs potentiation I do wonder whether the Zn potentiation is due to a second Zn site or is a special slow allosteric consequence of Zn inhibition dependent on the loop. Zn inhibition and potentiation both begin to occur at 0.3 mM. Perhaps the concentration response of inhibition vs potentiation by Cu could address the question from another angle. Two separate divalent sites might not have the same relative affinities for Zn and Cu.

It was difficult to tell if the description in methods of the kinetic model fully describes the model. There appear to be 24 different rate constants in the model. Perhaps labeling a table listing all the free parameters using a nomenclature clearly addressable to Figure 9A would help readers understand how the model works.

*Reviewer #3 (Recommendations for the authors):*

– Line 4 typo "vesibular".

– Figure 1B: typo in figure legend "Inhibtion".

– Figure 1B: mOTOP 3 currents: is there a reason for removing the Zn2+ before reaching a steady-state current (the current during Zn2+ application shows the sum of the two different actions but does it reach a steady-state)?

– Figure 5. I suppose that all the divalent ions tested show a block of OTOPs, although data are not shown. Is there a correlation between the potency of inhibition and the potency of activation for each of the ions? For instance, Cu2+ is the most potent activator among the tested ones, does it also block with the highest potency? It would be interesting if, among these transition metals, the authors could isolate one that has different affinities of inhibition/activation to find a concentration suitable to study the two modulation mechanisms independently.

– Figure 6 Tm11-12 linker: from the chicken OTOP3 structure this linker seems to be the longest on the extracellular side. To characterize the properties of this linker to the fullest the authors should test two important aspects: does the length of the linker affect Zn2+ potentiation? This would also partly answer the extent of the interaction of Zn2+ coordination with other parts of the channel. Obviously, a long linker has the flexibility to reach out to residues not in the closest proximity. The second aspect that should be worth testing is if the position of this linker, located between tm 11 and 12, is also important for Zn2+ potentiation. What would happen if the linker is moved around OTOP3 extracellular side? Does the Zn2+ effect correlate only with the linker itself (aminoacidic sequence) or even with its position in the protein? This could also provide insights into the gating mechanism itself.

– Figure 8. Given the position of the loop in the chicken OTOP3 structure (PDB 6NF6), supported by the quality of the electron density map, H531 could potentially flip towards H234 and E238 (mOTOP3 numbering). Those residues are both conserved in chicken and mouse isoforms, so it should be worth testing if they can be part of the zinc-binding site through site-directed mutagenesis, as their position is reasonably close to H531 and not too buried into the membrane.

– Figure 8 Given the position of E535A in the chicken OTOP3 structure I would suggest testing the E535H mutation, to measure possible enhancement of the Zn2+ potentiation.

– Figure 8 Given the effect of H531 in OTOP3 it would be reasonable to wonder what happens when the same position is replaced with a histidine in OTOP2 and OTOP1, both bearing an arginine. Is the histidine sufficient to recapitulate the O3/O2tm11-12 phenotype? Or at least part of it?

– Figure 8 Regarding the previous point: do the authors have an explanation for the OTOP1 potentiation given that this isoform does not have the corresponding H531 (OTOP3 numbering) residue?

– Line 301 "It is tempting to speculate that the binding site spans parts of the channel that are at a distance in the closed state and that come together in the open state as shown for other ion channels" (Gordon and Zagotta, 1995b; Zhang et al., 2009).

The Zn2+ potentiation occurs even during pre-treatment at pH 7.4, a condition in which the channel is closed (Teng et al. 2022). Doesn't this suggest that the binding site is available also during the closed state? The authors suggested direct competition with H^+^, so that at pH 7.4 Zn2+ can bind to the extracellular side (interacting with the unprotonated H531) transitioning the channel into a state that facilitates pore opening at pH 5.5. If this is the case what is the mechanism of Zn2+ binding when H531 is protonated (experiment at pH5.5 Figure 1B)? I think that this aspect of the mechanism is still a bit elusive and needs clarification.

---

## [Author Response]

Essential revisions:After consultation, the reviewers concur on these essential revisions to the manuscript.1) In the explanation of the model predictions, it is not clear how the allosteric model can predict the desensitization of currents. This part needs a more detailed explanation. Also, it was difficult to tell if the description in methods of the kinetic model fully describes the model. There appear to be 24 different rate constants. Perhaps labeling a table listing all the free parameters using a nomenclature clearly addressable to Figure 9A would help readers understand how the model works.

We apologize for not making the model parameters sufficiently clear. We have updated the methods section on the model to better describe the parameters and how they map onto the scheme in Figure 9A. We note that there are 14 free parameters in the model, 11 of which are likely to be of the most importance. We also note that we did not attempt to optimize the model parameters, in part because there are so many, but instead manually found a parameter set that qualitatively describes our observations. To further help illustrate how the model works, we have included two new Supplementary Figures. Supplementary Figure 2 shows the model’s simulated time course for each individual gating or binding domain, and Supplementary Figure 3 contrasts the simulated time course for the decay of the zinc-potentiated current with that of zinc unbinding from the activating site to show that they are similar (i.e., the current decay can be largely accounted for by zinc unbinding in the model simulation). We also added a few lines describing this conceptually in the model section of the main text.

2) Given the effect of H531 in OTOP3 it is reasonable to wonder if the histidine is sufficient to recapitulate the O3/O2tm11-12 phenotype or at least part of it. The same position should be replaced with histidine in OTOP2 and OTOP1, which have arginines. Related to this point: can the authors provide an explanation for the OTOP1 potentiation given that this isoform does not have the corresponding H531 residue?

The reviewers raise an interesting point. We have now tested the effect of introducing the equivalent residue to H531 in OTOP2 and OTOP1 (new Figure 8, supplementary figure 1). In the case of OTOP2, the channels remain insensitive to potentiation from Zn^2+^. Thus, H531 is not sufficient in the context of OTOP2 to recapitulate the effect of swapping the entire 11-12 linker. We assume that other residues in the 11-12 linker, or changes in the overall conformation of the linker, are required to confer sensitivity to Zn^2+^. For OTOP1, we did observe an enhancement in the degree of potentiation for channels with H replacing R, although the channels are not as strongly potentiated as OTOP3 channels. This data is consistent with the notion that the Zn^2+^ binding site consists of several residues and that their relative contributions to energetically stabilizing Zn^2+^ in the open/potentiated state may vary among different channels. It also is consistent with our model, which predicts that for a channel that is already activated by H^+^ (as OTOP1 is at the pH tested), Zn^2+^ will have a relatively smaller effect. There is certainly more to be done in determining the nature of the Zn^2+^ activating site, in closed and open conformations of the channels, and within the context of different OTOP isoforms.

3) Please comment on to what extent the desensitization of the proton currents could be a reflection of proton depletion given the large amplitude of the proton currents.

The decay of the currents was not a focus of this manuscript. It likely includes contributions from three factors: Zn2+ unbinding (de-potentiation), channel desensitization and ion accumulation (collapse of the H^+^ gradient). In a separate manuscript, Teng et al., 2022a, we discuss this question in the context of OTOP1 and find that the decay of the currents is likely due to collapse of the H^+^ gradients. For the experiments shown with OTOP3, our modeling shows that the decay of the currents can be be explained by unbinding of Zn2+. However, we cannot rule out that there is a component of either desensitization or gradient collapse. We now discuss this point when we describe the model.

Reviewer #1 (Recommendations for the authors):Specific comments:1) In several places in the manuscript, the authors mention that Zn potentiation indicates that the channels are gated, however, isn't activation by H an indication that the channels are are gated? Perhaps this statement should be explained in a more nuanced manner.

Wording changed throughout. Line 58; wording changed to indicate that evidence that channels are gated is confirmatory.

2) The solutions employed have a relatively low concentration of buffer, thus the proton currents will incur in proton depletion artifacts, especially for large amplitude currents. Please comment on to what extent the desensitization of the proton currents could be a reflection of proton depletion.

The buffer concentration was 10 mM, which is standard for this type of experiment. Higher concentrations did not change the kinetics of decay of other OTOP channels, in our hands, so we did not try higher concentrations for these experiments. The decay of the currents in these experiments is likely due to unbinding of Zn2+ and recovery of channels from the potentiated state (see essential revision #1 above).

3) The data in figure 3A shows the time course of the wash-off of the Zn effect. What process do the authors think is reflected here? A slow k_off_ for Zn unbinding or a slow rate constant for recovery from the potentiated state?

For the model in Figure 9A these are one and the same as the potentiated state is defined as that having Zn bound to the activating/potentiating site. This model, and thus Zn unbinding, can qualitatively explain the observed wash-off time course. We did not attempt to experimentally distinguish between them so we cannot provide a test of this hypothesis with available data.

4) In the methods section, please explain in more detail the procedure to perfuse cells in whole-cell recordings. Were the cells lifted and brought to the perfusion tubes? Or were the cells attached and the surrounding solution changed? If the latter, how complete is the solution exchange to be expected?

We apologize for leaving out this important detail. The cells were lifted. In similar experiments (Teng et al., 2022), we showed that complete solution exchange takes around 30 ms. This is now explained in the methods, Line 382.

5) In the explanation of the model predictions, it is not clear to me how the allosteric model can predict the desensitization of currents. This part needs a more detailed explanation.

Please see our response to essential revision #1.

Reviewer #2 (Recommendations for the authors):It was curious that it didn't seem like potentiation could occur in the presence of Zn. Despite the identification of an important His residue and the different concentration response of inhibition vs potentiation I do wonder whether the Zn potentiation is due to a second Zn site or is a special slow allosteric consequence of Zn inhibition dependent on the loop. Zn inhibition and potentiation both begin to occur at 0.3 mM. Perhaps the concentration response of inhibition vs potentiation by Cu could address the question from another angle. Two separate divalent sites might not have the same relative affinities for Zn and Cu.

Actually, we do observe potentiation in the presence of Zn, see Figure 1 C, where one can see a recovery of the currents following an initial rapid block. In the manuscript, we presented several pieces of evidence showing that the Zn blocking site is distinct from the Zn potentiating site. This includes: (1) a difference in dose-dependence shown in Figure 1D, where it can be seen that inhibiton saturates at 3 mM Zn, while activation does not saturate even at 10 mM, implying that the inhibitory site is higher affinity, (2) the observation that all OTOP channels are blocked by Zn but only OTOP3, and to a lesser extent OTOP1, is potentiated and (3) the observation that potentiation can be eliminated in the OTOP3/OTOP2 11-12 chimera, which retains inhibition.

We followed the reviewer’s suggestion to look a little more carefully at inhibition by other transition metals that we found could activate mOTOP3 channels. Thus, we now include new data addressing this point. This data, now shown in Figure 5, shows that Ni, Cd, and Cu all block mOTOP3 channels. Interestingly, Ni produces nearly identical block of the channels as Zn, but much less potentiation, consistent with different metal binding affinities for the two sites.

It was difficult to tell if the description in methods of the kinetic model fully describes the model. There appear to be 24 different rate constants in the model. Perhaps labeling a table listing all the free parameters using a nomenclature clearly addressable to Figure 9A would help readers understand how the model works.

Please see our response to essential revision #1.

Reviewer #3 (Recommendations for the authors):– Line 4 typo "vesibular".

Fixed

– Figure 1B: typo in figure legend "Inhibtion".

Fixed

– Figure 1B: mOTOP 3 currents: is there a reason for removing the Zn2+ before reaching a steady-state current (the current during Zn2+ application shows the sum of the two different actions but does it reach a steady-state)?

The duration of the Zn2+ stimulus in these experiments was based on the time needed to observe its inhibitory effect and we did not attempt to measure the steady-state activating effects under these conditions. This was done more carefully when we switched to a protocol that allowed us to study activation in the absence of inhibition.

– Figure 5. I suppose that all the divalent ions tested show a block of OTOPs, although data are not shown. Is there a correlation between the potency of inhibition and the potency of activation for each of the ions? For instance, Cu2+ is the most potent activator among the tested ones, does it also block with the highest potency? It would be interesting if, among these transition metals, the authors could isolate one that has different affinities of inhibition/activation to find a concentration suitable to study the two modulation mechanisms independently.

This is an excellent point. As described in the response to Reviewer 2, we now show that Ni, Cd and Cu (1 mM) all block mOTOP3 currents (pH 5.5). Thus, we could not find a metal ion that activates the currents without blocking them. There was no direct relationship between potency of inhibition and activation, and Ni which produced very little activation was as good a blocker as Zn.

– Figure 6 Tm11-12 linker: from the chicken OTOP3 structure this linker seems to be the longest on the extracellular side. To characterize the properties of this linker to the fullest the authors should test two important aspects: does the length of the linker affect Zn2+ potentiation? This would also partly answer the extent of the interaction of Zn2+ coordination with other parts of the channel. Obviously, a long linker has the flexibility to reach out to residues not in the closest proximity. The second aspect that should be worth testing is if the position of this linker, located between tm 11 and 12, is also important for Zn2+ potentiation. What would happen if the linker is moved around OTOP3 extracellular side? Does the Zn2+ effect correlate only with the linker itself (aminoacidic sequence) or even with its position in the protein? This could also provide insights into the gating mechanism itself.

The tm11-12 linkers are exactly the same length for mOTOP2 and mOTOP3, so length of the linker cannot explain the difference in sensitivity to Zn2+ potentiation. We have no doubt that changing the length of the linker could change function, but it is not clear what one could conclude from this experiment. Similarly, we do not propose that the linker on its own, irrespective of protein context, would potentiate the channels, and thus, moving the linker or swapping linkers would be extremely unlikely to open (potentiate) the channels.

– Figure 8. Given the position of the loop in the chicken OTOP3 structure (PDB 6NF6), supported by the quality of the electron density map, H531 could potentially flip towards H234 and E238 (mOTOP3 numbering). Those residues are both conserved in chicken and mouse isoforms, so it should be worth testing if they can be part of the zinc-binding site through site-directed mutagenesis, as their position is reasonably close to H531 and not too buried into the membrane.

Excellent point. We tested mutations of H234 and E238 to alanine and found they reduced potentiation. A double mutation had a more severe effect (although it did not completely eliminate potentiation like H531A) which suggests that the four residues may constitute the Zn2+ (metal ion) potentiating site. We now include this data in a new figure 8. Future structural studies will be necessary to confirm that metal ions bind to this site and to understand and how metal binding stabilizes an open state.

– Figure 8 Given the position of E535A in the chicken OTOP3 structure I would suggest testing the E535H mutation, to measure possible enhancement of the Zn2+ potentiation.

Indeed, this might be an interesting mutant to test, but we do not believe the results, however it would turn out, would impact the conclusion of our study, and for this reason we did not prioritize is among the many mutants we could generate and test.

– Figure 8 Given the effect of H531 in OTOP3 it would be reasonable to wonder what happens when the same position is replaced with a histidine in OTOP2 and OTOP1, both bearing an arginine. Is the histidine sufficient to recapitulate the O3/O2tm11-12 phenotype? Or at least part of it?

See responses to essential reviews.

– Figure 8 Regarding the previous point: do the authors have an explanation for the OTOP1 potentiation given that this isoform does not have the corresponding H531 (OTOP3 numbering) residue?

This is an interesting question. We did find that mutation to H increased the degree of potentiation, but it is still curious that the positive charge on the arginine would not disrupt potentiation in the context of mOTOP1 as it does in the context of mOTOP3.

– Line 301 "It is tempting to speculate that the binding site spans parts of the channel that are at a distance in the closed state and that come together in the open state as shown for other ion channels" (Gordon and Zagotta, 1995b; Zhang et al., 2009).The Zn2+ potentiation occurs even during pre-treatment at pH 7.4, a condition in which the channel is closed (Teng et al. 2022). Doesn't this suggest that the binding site is available also during the closed state?

There are two mechanisms by which Zn2+ could bind to the channel at pH 7.4. It could either bind to the closed state (which is something that our model predicts can happen at a low rate) or the channel can briefly transit through an open state (at low probability) from which Zn2+ will more rapidly bind. In either case, unbinding is predicted to be slow. This is consistent with the slow apparent on rate of Zn2+ at pH 7.4 (many seconds) and slow off rate (>1 s). Essentially, Zn2+ traps the channel in an open state.

The authors suggested direct competition with H^+^, so that at pH 7.4 Zn2+ can bind to the extracellular side (interacting with the unprotonated H531) transitioning the channel into a state that facilitates pore opening at pH 5.5. If this is the case what is the mechanism of Zn2+ binding when H531 is protonated (experiment at pH5.5 Figure 1B)? I think that this aspect of the mechanism is still a bit elusive and needs clarification.

For the experiment where the pH is 5.5, and the His residues is protonated, we need to presume that Zn2+ can bind, perhaps weakly, due to interactions with other residues, and that binding of Zn2+ causes the deprotonation of HIs531, which then may stabilize the Zn2+ ion. This is speculation, and we agree with the reviewer that there is still some question regarding the exact physical mechanism by which Zn binds to closed vs. open states of the channels and stabilizes the open state. We did not presume to have defined the precise role of H531 and other residues in this process, which may require structural information.